# SpikingMamba: Towards Energy-Efficient Large Language Models via Knowledge Distillation from Mamba

**Yulong Huang**[*]                                          *yhuang496@connect.hkust-gz.edu.cn*
*The Hong Kong University of Science and Technology (Guangzhou)*

**Jianxiong Tang**[*]                                          *jiatang@cityu.edu.hk*
*Department of Computer Science, City University of Hong Kong*

**Chao Wang**[*]                                          *wangc2023@mail.sustech.edu.cn*
*Department of Computer Science and Engineering, Southern University of Science and Technology, Shenzhen*

**Ziyi Wang**                                          *51265901129@stu.ecnu.edu.cn*
*School of Computer Science and Technology, East China Normal University, Shanghai*

**Jianguo Zhang**                                          *zhangjg@sustech.edu.cn*
*Department of Computer Science and Engineering, Southern University of Science and Technology, Shenzhen*

**Zhichao Lu**                                          *zhichao.lu@cityu.edu.hk*
*Department of Computer Science, City University of Hong Kong*

**Bojun Cheng** ✉                                          *bocheng@hkust-gz.edu.cn*
*The Hong Kong University of Science and Technology (Guangzhou)*

**Luziwei Leng** ✉                                          *lengluziwei@huawei.com*
*ACSLab, Huawei Technologies Co., Ltd., Shenzhen*

[*]These authors contributed equally. ✉Corresponding authors.
YL Huang did this work during an internship at Huawei ACSLab.

**Reviewed on OpenReview:** https://openreview.net/forum?id=uxb2jcCLxt&referrer

## Abstract

Large Language Models (LLMs) have achieved remarkable performance across tasks but remain energy-intensive due to dense matrix operations. Spiking neural networks (SNNs) improve energy efficiency by replacing dense matrix multiplications with sparse accumulations. Their sparse spike activity enables efficient LLMs deployment on edge devices. However, prior SNN-based LLMs often sacrifice performance for efficiency, and recovering accuracy typically requires full pretraining, which is costly and impractical. To address this, we propose SpikingMamba, an energy-efficient SNN-based LLMs distilled from Mamba that improves energy efficiency with minimal accuracy sacrifice. SpikingMamba integrates two key components: (a) SI-LIF, a signed-integer spiking neuron that preserves semantic polarity through signed multi-level spike representations. (b) A training-exclusive Smoothed Gradient Compensation (SGC) path mitigating quantization loss while preserving spike-driven efficiency. We employ a single-stage distillation strategy to transfer the zero-shot ability of pretrained Mamba and further enhance it via reinforcement learning (RL). Experiments show that SpikingMamba-1.3B achieves a $4.76\times$ energy benefit, with only a 4.78% zero-shot accuracy gap compared to the original Mamba. The model achieves a further 2.55% accuracy improvement after RL, narrowing the performance gap from 4.78% to 2.23%.

# 1 Introduction

Large language models (LLMs) have demonstrated remarkable performance across a wide range of tasks and domains (Li et al., 2025; Jiang et al., 2023; Liu et al., 2024; Team et al., 2023; Touvron et al., 2023; Chen et al., 2024a; Geiping et al., 2025; Jimenez et al., 2023; Wang et al., 2024b). However, their computational and energy requirements grow rapidly with scale, limiting their deployment in latency and energy constrained environments (Argerich & Patiño-Martínez, 2024).

These limitations stem largely from the fact that most modern LLMs are built on decoder-only Transformer architectures, where the self-attention mechanism incurs quadratic $O(L^2)$ complexity during inference. Transformer-based LLMs execute inference with a prefill stage followed by an autoregressive decoding stage (Touvron et al., 2023). In both stages, the self-attention mechanism suffers from quadratic $O(L^2)$ complexity with respect to sequence length $L$ (Yang et al., 2023). Additionally, the autoregressive decoding requires maintaining a growing Key-Value (KV) cache during inference, leading to $O(L)$ memory overhead. These limitations lead to inefficient computation, especially for long-context inference.

To address the quadratic cost of attention, the recently proposed Mamba architecture (Gu & Dao, 2023; Dao & Gu, 2024) replaces attention with a selective state space model (SSM). This design achieves linear-time $O(L)$ sequence modeling while requiring only $O(1)$ memory during autoregressive generation (Wang et al., 2024a), eliminating KV caching and improving efficiency for long-context and edge scenarios. However, despite these architectural improvements, Mamba still relies heavily on dense matrix multiplications, making energy consumption a critical bottleneck, especially on battery-powered or embedded devices (Xing et al., 2025).

To address this challenge, we propose **SpikingMamba**, a spiking large language model that combines the architectural efficiency of Mamba with the energy-saving advantages of SNNs. Rather than training a spiking model from scratch, we distill SpikingMamba from a pretrained Mamba2 using a single-stage self-distillation strategy. However, naive LIF neurons struggle to preserve semantic information during distillation, leading to degraded feature representations and suboptimal performance. To overcome this, we design a SI-LIF neuron and a Smoothed Gradient Compensation path for SpikingMamba modeling. Together, these components improve distillation quality while maintaining the sparse, low-power nature of SNNs. **Our contributions are summarized**:

- We propose SpikingMamba, a recurrent spiking LLMs that integrates a novel Signed Integer Integrate-and-Fire (SI-LIF) neuron. This design enhances feature representation for distillation by introducing negative activations, forming a ternary spiking scheme that better captures the magnitude and polarity of semantic representations.

- We introduce the Smoothed Gradient Compensation (SGC) path to mitigate quantization-induced representational fidelity loss from spikes. This auxiliary path operates exclusively during training, preserving spike-driven inference advantages. When applied to just 3 layers, SGC path improves 1.3B SpikingMamba's zero-shot accuracy by 0.8% on average.

- We propose Single-Stage Self-Distillation strategy, combining KL divergence on logits with hidden-state alignment loss, effectively transferring zero-shot ability from Mamba without full pretraining. Additionally, SpikingMamba is compatible with RL methods such as DPO (Rafailov et al., 2024) and KTO (Ethayarajh et al., 2024), which further enhance performance with tiny cost.

- In experiments on a 1.3B-parameter model, SpikingMamba achieves a ~4.76× energy benefit over Mamba2, while maintaining commonsense zero-shot accuracy within a 4.78% degradation margin after distillation. With reinforcement learning, SpikingMamba achieves a further 2.55% accuracy improvement, using fewer than 10 GPU-hours.

## 2 Related Work

### 2.1 Mamba and Distillation.

The Mamba (Gu & Dao, 2023) architecture and its successor Mamba2 (Dao & Gu, 2024) have been proposed as efficient recurrent alternatives to Transformers for sequence modeling. Leveraging structured state space models (SSMs), they eliminate $O(L^2)$ quadratic attention and $O(L)$ KV caching, while achieving performance on par with or better than Transformers of similar size (Wang et al., 2024a). Their recurrent structure and constant memory make Mamba-based LLMs well-suited for edge deployment. Therefore, recent studies such as MambaInLLaMA (Wang et al., 2024a), LoLCats (Zhang et al., 2024) and Llamba (Bick et al., 2025) explore distilling Transformer-based LLMs into Mamba-based models with minimal training cost.

### 2.2 SNNs for LLMs and Mamba

Improving LLM efficiency remains a fundamental challenge for edge deployment (Qu et al., 2025; Cai et al., 2024). Quantization reduces model size and compute, yet activation quantization is difficult due to high-magnitude outliers in LLMs, often inducing severe accuracy loss under low-bit settings (Lin et al., 2024; Yu et al., 2025). Additionally, quantization does not alleviate the inherent energy cost of dense matrix multiplications. In contrast, spiking neural networks (SNNs) utilize binary, event-driven activations that exhibit both spatial and temporal sparsity. For example, prior work reports spike rates as low as 20% (Tang et al., 2024a), this sparsity translates to significant savings in computation and I/O bandwidth, replacing dense MAC operations with sparse Accumulations (AC). These properties make SNNs a promising direction for energy-efficient inference.

Recent work on spiking language models follows two main paradigms: converting pretrained ANNs or training SNNs from scratch. Conversion-based methods, such as SpikingBERT (Bal & Sengupta, 2024), Spike-BERT (Lv et al., 2023), and SpikeLLM (Xing et al., 2025), reuse pretrained weights but require many inference timesteps for spike accumulation, increasing latency and energy cost. Alternatively, models like SpikeGPT (Zhu et al., 2023), SpikingSSMs (Shen et al., 2025), and SpikeLM (Xing et al., 2024b) are trained from scratch and support sparse computation, but incur high training costs to scale larger models. Table 1 compares these models with different training strategies, neuron step, inference complexity and zero-shot ability.

Recent efforts have explored combining SNNs with Mamba, primarily in vision tasks such as video understanding (Li et al., 2024), event-based classification (Qin & Liu, 2024), action recognition (Chen et al., 2024b), and point cloud processing (Wu et al., 2025). These methods are vision-centric and typically replace the activation function with spiking neurons or stack Mamba blocks alongside SNN modules. However, Mamba was originally designed for language modeling, and its integration with SNNs for energy-efficient LLMs remains unexplored.

Table 1: Comparison of SNN-based language models. SpikingMamba is the first linear-time SNN LLM at the billion-parameter scale supporting distillation, low-latency inference, and reinforcement learning. $T$ denotes the number of repeated computations per token and $D$ denotes an integer range. A larger $T$ leads to higher inference latency.

| Model | Size | Training | Step ($T \times D$) | Infer. Comp. | Zero-Shot |
|---|---|---|---|---|---|
| SpikeBERT (Lv et al., 2023) | 109 M | 2-stage distill | $4 \times 1$ | $O(4L^2)$ | ✗ |
| SpikingBERT (Bal & Sengupta, 2024) | 50 M | 3-stage distill | $125 \times 1$ | $O(125L^2)$ | ✗ |
| SpikeLM (Xing et al., 2024b) | 194 M | 2-stage distill | $4 \times \pm 1$ | $O(4L^2)$ | ✗ |
| SpikeGPT (Zhu et al., 2023) | 216 M | Direct + Finetune | $1 \times 1$ | $O(L)$ | ✗ |
| SpikingSSMs (Shen et al., 2025) | 75 M | Direct | $1 \times 1$ | $O(L)$ | ✗ |
| SpikeSSMs (Zhong et al., 2024) | 75 M | Direct | $1 \times 1$ | $O(L)$ | ✗ |
| SpikeLLM (Xing et al., 2024a) | 7 B - 70 B | Quantization | $2 \times 16$ | $O(32L^2)$ | ✓ |
| **SpikingMamba (Ours)** | 1.3 B | 1-stage distill + RL | $1 \times \pm 4$ | $O(4L)$ | ✓ |

## 3 Preliminary and Motivation

**Notation.** Throughout the paper and Appendix, vectors and matrices are denoted by bold italic lowercase and bold capital letters, respectively (e.g., $\boldsymbol{x}$ and $\boldsymbol{W}$). For a fully connected layer, $\boldsymbol{y} = \boldsymbol{W}\boldsymbol{x} \in \mathbb{R}^{d_{\text{out}}}$, where $\boldsymbol{W} \in \mathbb{R}^{d_{\text{out}} \times d_{\text{in}}}$ and $\boldsymbol{x} \in \mathbb{R}^{d_{\text{in}}}$. The symbol $\boldsymbol{W}_{:,i}$ denotes the $i$-th column of matrix, and $\left(\sum_{\{i|\text{condition}\}} \boldsymbol{W}_{:,i}\right)$ indicates the column-wise sum over indices satisfying the condition. The subscript $\boldsymbol{x}_t$ refers to the variable $\boldsymbol{x}$ at the $t$-th token, while the bracket $\boldsymbol{x}[t]$ denotes the $t$-th micro-timestep within neuron dynamics per token.

### 3.1 Spiking Neurons

The Leaky Integrate-and-Fire (LIF) is one of the most widely adopted neuron models for modeling spiking behaviors in SNNs (Huang et al., 2024b; Meng et al., 2023). At each timestep $t$, the neuron at layer $l$ integrates its postsynaptic current $\boldsymbol{c}^l[t]$ with its membrane potential from the previous timestep $\boldsymbol{u}^l[t-1]$ (Eq 1):

$$\boldsymbol{u}^l[t] = \beta(\boldsymbol{u}^l[t-1] - V_{\text{th}}\boldsymbol{s}^l[t-1]) + \boldsymbol{c}^l[t], \tag{1}$$

$$\boldsymbol{s}^l[t] = \Theta(\boldsymbol{u}^l[t] - V_{\text{th}}), \tag{2}$$

where $\beta \in (0,1)$ is the decay factor mimicking the leaky mechanism. The postsynaptic current $\boldsymbol{c}^l[t]$ is computed by the synaptic operation $*$ (either fully connected or convolutional) between the weight matrix $\boldsymbol{W}^l$ and the presynaptic spikes $\boldsymbol{s}^{l-1}[t]$ (i.e., $\boldsymbol{c}^l[t] = \boldsymbol{W}^l * \boldsymbol{s}^{l-1}[t]$). When the membrane potential $\boldsymbol{u}^l[t]$ exceeds a threshold $V_{\text{th}}$, the neuron emits a binary spike $\boldsymbol{s}^l[t] \in \{0,1\}$ according to the Heaviside step function $\Theta(x)$, which outputs 1 for $x \geq 0$ and 0 otherwise, as shown in Eq 2.

To address the quantization errors inherent in standard LIF neurons, Luo et al. (2024) proposed the Integer Leaky Integrate-and-Fire (I-LIF) neuron. I-LIF emits integer-valued outputs during training and converts them into binary (0/1) spikes during inference, thereby enhancing training dynamics while preserving spike-driven inference. Specifically, the spiking function in Eq 2 is modified as:

$$\boldsymbol{s}^l[t] = \text{Clip}\left(\text{Round}(\boldsymbol{u}^l[t]), 0, D\right), \tag{3}$$

where $\text{Round}(\cdot)$ denotes the rounding function, $\text{Clip}(x, \min, \max)$ restricts $x$ to the range $[\min, \max]$, and $D$ is a hyperparameter indicating the maximum integer output. By producing discrete integer outputs during training (Yao et al., 2025), I-LIF enhances optimization performance while maintaining efficient 0/1 spike-driven inference at test time, as demonstrated by Luo et al. (2024). The concurrent work SpikingBrain (Pan et al., 2025) produces integer-valued activations for efficient forward inference. However, this work is designed exclusively for quantized inference and does not verify end-to-end training.

### 3.2 Motivation

In Mamba2, over 90% of parameters (e.g., 92.3% in 1.3B) are in the input/output Linear layer, corresponding to $\boldsymbol{W}_{\text{in}}$ and $\boldsymbol{W}_{\text{out}}$ respectively (Further details are provided in the Appendix A and B). The embedding layer shares parameters with the causal head, but its relative size shrinks as the model scales. Binarizing embeddings harms token representation and leads to performance loss. Therefore, we only focus on the Linear layer due to the computational dominance of the Linear layer in Multiply-Accumulate (MAC) operations. To reduce this computational cost, we insert spiking neurons upstream of input and output projections, converting dense MACs into sparse Accumulation (AC) operations. The Sparse AC operations refer to accumulation operations without the multiplication step. This occurs in event-driven spiking computation, where additions are performed only when spikes are present, therefore leading to sparsity in the operation count. This design enhances energy efficiency while preserving semantic fidelity.

## 4 Method

We propose SpikingMamba, an energy-efficient extension of Mamba2 designed to maintain performance with reduced computational cost. It introduces three novel components: (i) a signed spiking neuron (SI-LIF) that

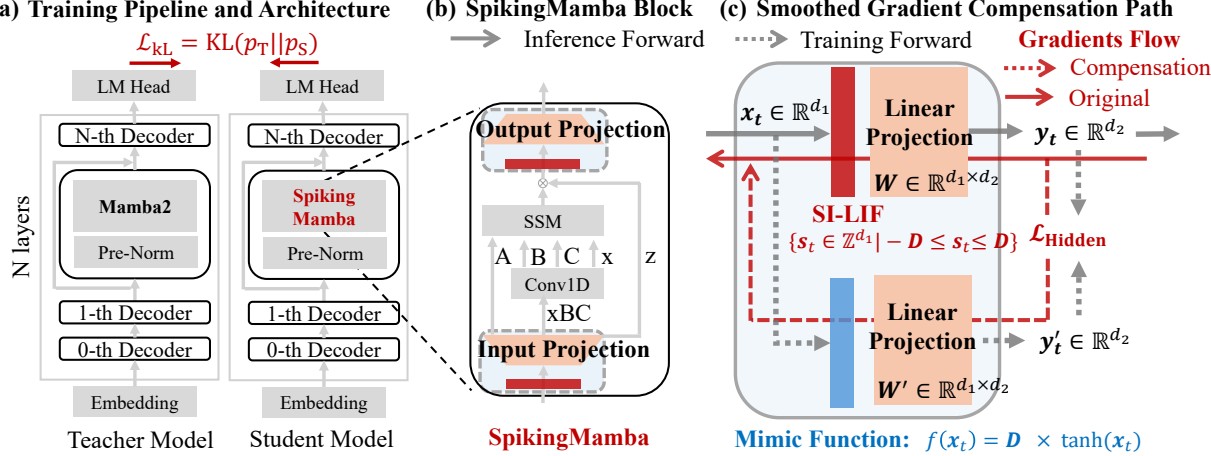

Figure 1: (a) Overview of the training architecture. (b) Illustration of the SpikingMamba block. (c) Illustration of the Smoothed Gradient Compensation path.

captures negative polarity and preserves magnitude during training; (ii) a lightweight Smoothed Gradient Compensation path that restores essential features lost in the spiking conversion, and (iii) a hybrid training strategy combining knowledge distillation and to transfer teacher model ability while reinforcement learning for improving model performance in further. An overview of the full framework is shown in Figure 1(a).

## 4.1 SpikingMamba Block

To reduce energy consumption in the most computationally expensive parts of the model, SpikingMamba introduces neurons before both the input and output projection layers. These projections dominate memory and compute overhead, making them ideal targets for spike-driven optimization.

At each timestep $t$, the input $\boldsymbol{x}[t] \in \mathbb{R}^{d_{\text{in}}}$ is passed through a spiking neuron function $f_{\text{SN}}(\cdot)$, yielding a sparse binary activation $\boldsymbol{s}[t] = f_{\text{SN}}(\boldsymbol{x}[t]) \in \{0,1\}^{d_{\text{in}}}$. Rather than performing dense matrix-vector multiplication $\boldsymbol{y}[t] = \boldsymbol{W}\boldsymbol{x}[t]$, where $\boldsymbol{W} \in \mathbb{R}^{d_{\text{out}} \times d_{\text{in}}}$ is the projection weight, we compute sparse row-wise accumulation over firing indices $i$:

$$\boldsymbol{y}[t] = \sum_{\{i|\boldsymbol{s}[t]_i=1\}} \boldsymbol{W}_{:,i} \quad , \tag{4}$$

where the conditional summation is determined by the sparse binary activation $\boldsymbol{s}[t] = f_{\text{SN}}(\boldsymbol{x}[t])$, which is produced by the spiking neuron and thus remains implicitly dependent on the input $\boldsymbol{x}[t]$. This converts costly MAC operations into efficient AC operations. The sparse spiking pattern further reduces memory traffic, improving overall compute and I/O efficiency.

### 4.1.1 SI-LIF Neuron

Existing integer spiking neurons remain mechanistically incomplete for large language models: I-LIF's strictly non-negative $[0, D]$ range erases semantic polarity (Luo et al., 2024), whereas ternary neurons (Guo et al., 2024; Xing et al., 2024b) with ($\{-1, 0, 1\}$) impose coarse magnitude discretizations, jointly provoking a distribution shift that permeates Mamba's recurrent state.

To rectify this dual defect, we propose the Signed Integer Leaky-Integrate-and-Fire (SI-LIF) neuron, a bounded signed-integer spike neuron that supports both integer-valued training and spike-driven inference.

### 4.1.2 Integer-Valued Training

At each step, the membrane potential is rounded and clipped into a tunable symmetric range:

$$s_t = \text{Clip}\big(\text{Round}(\boldsymbol{x}_t), -D, D\big), \tag{5}$$

here, $D$ explicitly determines the trade-off in quantization accuracy, while the gradient of $\boldsymbol{s}_t$ with respect to $\boldsymbol{x}_t$ is approximated using a rectangular surrogate function, which yields zero derivatives outside the interval:

$$\frac{\partial \boldsymbol{s}_t}{\partial \boldsymbol{x}_t} = \left\{ \begin{array}{ll} \alpha, & -D \leq \boldsymbol{x}_t \leq D, \\ 0, & \text{otherwise,} \end{array} \right. \tag{6}$$

where the scaling factor $\alpha$ denotes the slope of the surrogate gradient. In this work, we adopt $\alpha = 1$ by default. Compared to I-LIF and ternary neurons, SI-LIF encodes information with both polarity and amplitude, thereby enhancing the representational capacity of SNNs during training.

### 4.1.3 Spike-Driven Inference

This paragraph details how the trained SI-LIF neuron performs event-driven inference while maintaining exact equivalence to the dense transformation $\boldsymbol{y}_t = \boldsymbol{W}\boldsymbol{s}_t$, where $\boldsymbol{s}_t = \text{SI-LIF}(\boldsymbol{x}_t) \in [-D, D]$. To enable the event-driven inference, we reconstruct the integer $\boldsymbol{s}_t \in [-D, D]$ as binary $\boldsymbol{s}[i] \in \{0, 1\}$ through the neuronal dynamics of SI-LIF within the micro-timestep $i = 1$ of an internal temporal window of length $D$. For $i = 1, \ldots, D$, the neuronal dynamics strictly obey the discrete-time LIF equations:

$$\mathbf{v}[i] = \beta(\mathbf{v}[i-1] - V_{\text{th}} \cdot \mathbf{s}[i-1]) + \mathbf{x}[i], \tag{7}$$

$$\mathbf{s}[i] = \Theta\big(\mathbf{v}[i] - V_{\text{th}}\big), \tag{8}$$

where $\beta = 1$ for exact reconstruction and $\mathbf{s}[i] \in \{0, 1\}^{d_{\text{in}}}$ denotes the binary spike vector at micro-step $i$.

Specifically, the $\mathbf{x}[1] = |\boldsymbol{x}_t|$ at first micro-timestep while $\mathbf{x}[i] = 0$ for other $i$, where the $|\cdot|$ denotes the absolute value taken. Consequently, the summation of the spiking sequence equals the integer activation $\mathbf{s}_t$ produced by SI-LIF during training. The layer output is then recovered exactly by a sparse weighted summation over the spike train:

$$\boldsymbol{y}_t = \underbrace{\boldsymbol{W} \cdot f_{\text{SN}}(\boldsymbol{x}_t)}_{\text{For Training}} = \sum_{i=1}^{D} \boldsymbol{W} \cdot \Big(\text{sgn}(\boldsymbol{x}_t) \cdot \mathbf{s}[i]\Big) = \sum_{i=1}^{D} \sum_{j=1}^{d_{\text{in}}} \Big(\text{sgn}(\boldsymbol{x}_t)_j \boldsymbol{W}_{:,j}\Big) \cdot s[i]_j = \underbrace{\sum_{\{j|s[i]_j=1\}} \text{sgn}(\boldsymbol{x}_t)_j \cdot \boldsymbol{W}_{:,j}}_{\text{For Spike-Driven Inference}}, \tag{9}$$

where the binary spike $s[i]$ is obtained from the dynamic Eq 7-8, then s[i] serves as a selection signal to choose the corresponding $j$-th row of the $\boldsymbol{W}_{:,j}$ to summarize together. Further, the $\text{sgn}(\boldsymbol{x}_t)$ is the single-bit sign flag. This sign retrieval incurs zero additional cost: the polarity bit is obtained in hardware by a single XOR with the sign bit of the input, enabling instantaneous sign inversion via bit-flip without additional arithmetic operation.

Each inner summation corresponds to a row-wise addition triggered exclusively by sparsity active spikes, thereby replacing the dense MAC ($\boldsymbol{W}\boldsymbol{x}_t$) with at most $D$ times event-driven accumulations while maintaining exact equivalence. These spike-based computations enable fine-grained signed encoding while maintaining neuromorphic sparsity.

## 4.2 Smoothed Gradient Compensation Path

Although SI-LIF neurons capture information from negative activation domains, their quantized characteristics can introduce gradient approximation errors during backpropagation. To alleviate these issues, we introduce a Smoothed Gradient Compensation (SGC) path (denoted by dashed lines in Figure 1(c)), which approximates the output of the primary spiking pathway and facilitates model training through its differentiable nature.

Specifically, we apply a distributional alignment constraint to enforce semantic consistency. This means that ensuring both pathways produce similar predictive distributions despite the quantized nature of the SI-LIF outputs:

$$\mathcal{L}_{\text{Hidden}} = \frac{1}{2T} \sum_{t=1}^{T} \|\text{softmax}(\boldsymbol{y}_t) - \text{softmax}(\boldsymbol{y}'_t)\|_2^2, \tag{10}$$

where $\boldsymbol{y}_t = \boldsymbol{W} \cdot f_{\text{SN}}(\boldsymbol{x}_t)$ is the output from the spiking pathway, and $\boldsymbol{y}'_t = \boldsymbol{W}' \cdot f_m(\boldsymbol{x}_t)$ is the output from the SGC path. The projection matrix $\boldsymbol{W}' \in \mathbb{R}^{d_1 \times d_2}$ is initialized based on $\boldsymbol{W} \in \mathbb{R}^{d_1 \times d_2}$ and subsequently updated through training.

To ensure scale consistency between $f_m(\boldsymbol{x}_t)$ and the SI-LIF output $f_{\text{SN}}(\boldsymbol{x}_t)$, a range-preserving mimic function is introduced:

$$f_m(\boldsymbol{x}_t) = D \times \tanh(\boldsymbol{x}_t), \tag{11}$$

where $D$ is the spike amplitude hyperparameter of SI-LIF neurons. This formulation ensures strict alignment in dynamic range $[-D, D]$ with the spiking outputs, while enabling smooth gradients $\partial \mathcal{L}_{\text{Hidden}}/\partial \boldsymbol{x}_t$ to support parameter optimization. Upon completion of training, only the spiking pathway is retained for event-driven inference.

## 4.3 Training Framework

To enable stable and effective training of SpikingMamba without full-scale pretraining, we adopt a learning framework: (1) knowledge distillation (KD) from a pretrained Mamba2 teacher, and (2) alignment via reinforcement learning (RL). This combination transfers both logit-level behavior and internal representations to the spiking model while addressing its representation shift problem.

### 4.3.1 Distillation Stage

We distillate SpikingMamba via supervised fine-tuning using teacher-generated pseudo-labels. The training objective includes both output-level and hidden-level alignment:

$$\mathcal{L} = \mathcal{L}_{\text{KL}} + \mathcal{L}_{\text{Hidden}}, \tag{12}$$

The first term minimizes the KL divergence between the output distributions of teacher and student:

$$\mathcal{L}_{\text{KL}} = \frac{1}{T} \sum_{t=1}^{T} \text{KL}\big(p(\cdot|\hat{y}_{1:t}, x, \theta_T) || p(\cdot|\hat{y}_{1:t}, x, \theta_S)\big), \tag{13}$$

where $\theta_T$ and $\theta_S$ are the teacher and student parameters, respectively. The second term $\mathcal{L}_{\text{Hidden}}$ corresponds to the hidden state alignment defined in Eq 10, ensuring fidelity of intermediate representations. In practice in this work, we only incorporate the SGC path on 3 layers with the first layer, middle layer and last layer, which could balance the training efficiency and performance. Specifically, these correspond to the 1st, 12th, and 24th layers in the 130M model, and the 1st, 24th, and 48th layers in the 1.3B model, which contain 24 and 48 layers in total, respectively.

### 4.3.2 Alignment Stage

To further enhance zero-shot reasoning and preference alignment, we apply reinforcement learning using Direct Preference Optimization (DPO) (Rafailov et al., 2023) and its stabilized variant KTO (Ethayarajh et al., 2024). Given a prompt $x$ with preferred response $y_w$ and dispreferred response $y_l$, DPO maximizes:

$$\pi_\theta = \max_\theta \mathbb{E}_{(x,y_w,y_l) \sim \mathcal{D}} \log \sigma \left(\beta \cdot (f_w - f_l)\right), \tag{14}$$

where $f_w = \log \frac{p(y_w|x;\theta)}{p(y_w|x;\theta_T)}$, $f_l = \log \frac{p(y_l|x;\theta)}{p(y_l|x;\theta_T)}$, $\sigma$ is the sigmoid function and $\theta_T$ is the reference policy. While DPO is effective, it is unstable due to local gradient noise and harsh penalization.

KTO addresses these issues by replacing the pairwise structure with a global reward baseline. For a single labeled pair $(x, y)$, the reward is $r(x, y) = \beta \log \frac{p(y|x;\theta)}{p(y|x;\theta_T)}$, which is compared to a dataset-level baseline $z_{\text{ref}}$. The final KTO loss becomes:

$$\mathcal{L}_{\text{KTO}} = \mathbb{E}_{(x,y)\sim\mathcal{D}}\Big[w(y)\Big(1 - \sigma\left(s_y \cdot (r(x, y) - z_{\text{ref}})\right)\Big)\Big], \tag{15}$$

where $s_y \in \{+1, -1\}$ denotes preference labels, and $w(y)$ is a sample weighting function. This allows for efficient use of both paired and unpaired data while improving training stability.

## 5 Experiments

To evaluate the effectiveness, efficiency, and generalization capabilities of SpikingMamba, we conduct experiments across multiple settings (The detailed experimental setup is provided in the section 5.1). We first assess zero-shot accuracy on general reasoning tasks and measure generative performance using perplexity on standard language modeling benchmarks. Next, we analyze the energy efficiency of SpikingMamba under various configurations. Finally, ablation studies further validate the contribution of each component and the distillation pipeline.

### 5.1 Experimental Setup

**Implementation Details.** During distillation, we perform supervised fine-tuning on the GenQA (Chen et al., 2024c), InfinityInstruct (BAAI, 2024), and OpenHermes 2.5 (Teknium, 2023) datasets, following the single-epoch strategy used in MambaInLLaMA (Wang et al., 2024a). The teacher is a pretrained Mamba2 of the same size, and SpikingMamba is initialized with the corresponding Mamba2 weights. For reinforcement learning (RL), we apply Direct Preference Optimization (DPO) (Rafailov et al., 2024) or its variant KTO (Ethayarajh et al., 2024) using the UltraFeedback (Cui et al., 2023). All models are trained with the AdamW optimizer ($\beta = (0.9, 0.98)$) and a global batch size of 32, using linear warm-up for the first 1% of steps followed by cosine annealing. The sequence length is fixed at 2048 tokens, and the embedding layer remains frozen during training. All experiments are conducted using 8 NVIDIA A100 GPUs with BF16 precision. For the 1.3B model, distillation takes around 42 hours, and RL takes around 1 hour. We use a fixed learning rate of 1e-3 for distillation and 5e-6 for RL.

**Evaluation Datasets.** We utilize the open-source LM Evaluation Harness library (Gao et al., 2023) (from the `big-refactor` branch) to evaluate six standard tasks: BoolQ accuracy (Clark et al., 2019), PIQA accuracy (Bisk et al., 2020), HellaSwag (HS) normalized accuracy (Zellers et al., 2019), WinoGrande (WG) accuracy (Sakaguchi et al., 2021), ARC-Easy and ARC-Challenge (AE and AC) accuracy and normalized accuracy (Clark et al., 2018). Each task is evaluated by analyzing the probability assigned by the model to each potential answer choice. We also report perplexity on the WikiText-2 (Merity et al., 2016), C4 (Raffel et al., 2020), PTB (Marcus et al., 1993) dataset as an additional metric. Perplexity measures how well a probability model predicts a token, quantitatively measuring the model's generation power.

**Baseline.** We primarily compare SpikingMamba against the pretrained Mamba2 baseline. We further include spike-based language models such as SpikeLLM (Xing et al., 2024a), SpikingSSMs (Shen et al., 2025) and SpikeGPT (Zhu et al., 2023) for a comprehensive comparison. Although SNNs and quantization are orthogonal methods, spiking out can be viewed as a 1-bit activation quantization. Thus, we additionally compare against the 1-bit quantized Mamba (Tang et al., 2024b).

### 5.2 Main Results

#### 5.2.1 Effectiveness of SpikingMamba

As shown in Table 2, both SI-LIF and the SGC path are critical to retaining Mamba performance in the spiking setting. Specifically, using the SGC path consistently improves performance compared to variants without it, indicating its effectiveness in recovering information loss from the SNN path. Furthermore,

Table 2: Comparison with 1-bit Mamba quantization and SNN-based LLMs. We adopt a unified "step" notation $T \times D$ from Yao et al. (2025), where $T$ is the number of timesteps and $D$ the maximum integer activation ("$D = 1$" is LIF, "$D > 1$" is I-LIF (Yao et al., 2025), "$\pm D$" is SI-LIF). $T > 1$ implies extra T times repeated computation needed for per token. "**P.**" denotes parameter count, "**T.**" indicates training tokens, and "**PT**" marks pretrained models. "**Diff.**" shows average accuracy gap (Red: Performance drop compared to the ANN model). The **SGC** denotes whether to use the Smoothed Gradient Compensation path. While **SpikingMamba$_s$** denotes our fully SNN model, we highlight with Orange background color.

| Model | P. | T. (↓) | SNN | SGC | Step | Zero-shot Accuracy (%) (↑) | | | | | | | |
| | | | | | | BoolQ | PIQA | HS$_{Norm}$ | WG | AE | AC$_{Norm}$ | Avg. | Diff. |
|---|---|---|---|---|---|---|---|---|---|---|---|---|---|
| Mamba2 | 130M | - | PT | - | 1 | 55.63 | 64.85 | 35.13 | 52.17 | 47.31 | 24.40 | 46.58 | - |
| SpikingMamba$_s$ | 130M | 7 B | ✓ | ✗ | $1 \times \pm 4$ | 51.01 | 61.10 | 31.44 | 51.62 | 46.59 | 22.78 | 44.09 | -2.49 |
| SpikingMamba$_s$ | 130M | 7 B | ✓ | ✓ | $1 \times \pm 4$ | 53.27 | 62.19 | 31.32 | 51.78 | 45.96 | 22.27 | **44.47** | **-2.12** |
| Mamba2 | 1.3B | - | PT | - | 1 | 64.34 | 73.72 | 59.94 | 61.09 | 64.18 | 33.11 | 59.40 | - |
| Bi-LLM (Mamba) (Huang et al., 2024a) | 1.3B | 1260 B | ✗ | - | - | 40.10 | 55.40 | 29.60 | 50.70 | 30.60 | 21.80 | 38.03 | -21.36 |
| Bi-Mamba (Tang et al., 2024b) | 1.3B | 1260 B | ✗ | - | - | 62.00 | 69.20 | 43.10 | 53.70 | 43.90 | 24.40 | 49.38 | -10.01 |
| SpikingMamba$_s$ | 1.3B | 7 B | ✓ | ✗ | $1 \times 4$ | 59.45 | 68.61 | 44.89 | 54.70 | 61.83 | 29.78 | 53.21 | -6.19 |
| SpikingMamba$_s$ | 1.3B | 7 B | ✓ | ✗ | $1 \times \pm 4$ | 58.78 | 69.53 | 48.85 | 54.22 | 62.58 | 29.18 | **53.86** | **-5.54** |
| SpikingMamba$_s$ | 1.3B | 7 B | ✓ | ✓ | $1 \times \pm 4$ | 59.60 | 70.34 | 48.66 | 55.72 | 63.44 | 29.95 | **54.62** | **-4.78** |
| LLAMA1-7B | 7B | - | PT | - | 1 | 73.08 | 77.47 | 73.00 | 67.07 | 52.48 | 41.46 | 64.09 | - |
| SpikeLLM (Xing et al., 2024a)[2025' ICLR] | 7B | - | ✓ | - | $2 \times 16$ | 65.45 | 41.67 | 32.51 | 64.37 | 56.59 | 54.3 | **52.48** | -11.57 |
| Mamba2 + DPO | 1.3B | - | ✗ | - | 1 | 63.64 | 73.23 | 61.22 | 60.69 | 64.02 | 37.03 | 59.97 | - |
| SpikingMamba$_s$ + DPO | 1.3B | - | ✓ | ✗ | $1 \times \pm 4$ | 61.80 | 69.86 | 50.15 | 53.99 | 63.26 | 31.14 | 55.03 | -4.94 |
| SpikingMamba$_s$ + DPO | 1.3B | - | ✓ | ✓ | $1 \times \pm 4$ | 62.23 | 70.73 | 52.39 | 57.14 | 63.64 | 35.32 | **56.91** | **-2.49** |
| Mamba2 + KTO | 1.3B | - | - | - | 1 | 60.55 | 73.78 | 61.67 | 61.48 | 66.12 | 35.41 | 59.84 | - |
| SpikingMamba$_s$ + KTO | 1.3B | - | ✓ | ✗ | $1 \times \pm 4$ | 63.61 | 70.46 | 52.12 | 56.59 | 65.28 | 32.34 | 56.73 | -3.11 |
| SpikingMamba$_s$ + KTO | 1.3B | - | ✓ | ✓ | $1 \times \pm 4$ | 62.72 | 71.33 | 52.20 | 56.27 | 66.29 | 34.22 | **57.17** | **-2.23** |

the proposed SI-LIF neuron, which supports both positive and negative spike amplitudes, outperforms the I-LIF variant that only produces positive spikes, demonstrating the benefit of richer activation dynamics. Compared to 7B SpikeLLM, 1.3B SpikingMamba exhibits a 4.78% performance drop in zero-shot accuracy while SpikeLLM-7B drops 11.57%. Importantly, SpikingMamba-1.3B achieves 54.62% zero-shot performance with just 7B tokens for distillation, while Bi-Mamba-1.3B achieves 49.38% in contrast to the 1280B tokens needed. Under the same training conditions, the SpikingMamba from 54.62% improves to 56.91% and 57.17% after DPO and KTO reinforcement learning, achieving 2.29% and 2.55% improvement, while Mamba2 improves only 0.57% and 0.44%, which indicates the SNN-based LLM could potentially benefit more from reinforcement learning than the ANN-based LLM.

### 5.2.2 Comparison to SNNs Training from Scratch

Table 3 further evaluates the zero-shot generalization ability of SpikingMamba against SNN-based language models trained from scratch. Despite not being specifically tuned on WikiText datasets, SpikingMamba achieves ($T \times D = 1 \times \pm 4$) lower perplexity than fully-trained SNN language models such as SpikeGPT and SpikingSSMs. This shows that by leveraging a pretrained teacher and an efficient distillation process, SpikingMamba can preserve zero-shot performance at a significantly lower training cost.

### 5.2.3 Perplexity

We analyze the perplexity (The lower is better) across datasets, including WikiText-2 (Merity et al., 2016), C4 (Raffel et al., 2020), PTB (Marcus et al., 1993) as done in Bi-Mamba (Tang et al., 2024b). We observe a consistent performance hierarchy across configurations (SI-LIF > I-LIF (Yao et al., 2025) > LIF) as shown in Table 4. This trend highlights the importance of ternary spikes for SNN-based LLMs modeling. The same trend also persists at the 130M scale (Further details are provided in the Supplementary Materials). While Bi-Mamba requires 1260B tokens and a larger teacher model, our method achieves comparable or better perplexity (e.g., 25.11 vs. 28.9 on PTB) using only 7B tokens and the same model size, demonstrating the efficiency of our distillation approach.

Table 3: Comparison with spiking language models on WikiText-103, measured in token-level perplexity (PPL) (↓). SpikeGPT (Zhu et al., 2023) and SpikingSSMs (Shen et al., 2025) are trained on the WikiText datasets, while SpikingMamba reports zero-shot results without any additional training or finetuning on these datasets.

| Model | SNN | Parameters | WikiText-103 PPL (↓) |
|---|---|---|---|
| GPT2 Small (Radford et al., 2019) | ✗ | 124M | 29.41 |
| GPT2 Medium (Radford et al., 2019) | ✗ | 346M | 26.37 |
| Mamba2 (Dao & Gu, 2024) | ✗ | 130M | 19.56 |
| SpikeGPT (Zhu et al., 2023) | ✓ | 216M | 39.75 |
| SpikingSSMs (Shen et al., 2025) | ✓ | 75M | 33.94 |
| SpikeSSMs (Zhong et al., 2024) | ✓ | 75M | 33.18 |
| **SpikingMamba (Ours)** | ✓ | 130M | **26.32** |

Table 4: The perplexity under the different configurations. "P." means the model parameters. "SGC" denotes using the Smoothed Gradient Compensation path. "Step" means a different neuron ("$D > 1$" is I-LIF (Yao et al., 2025), "$\pm D$" is SI-LIF).

| | Parameters | SGC | Step | Wiki2 (↓) | C4 (↓) | PTB (↓) |
|---|---|---|---|---|---|---|
| **Mamba2** | 1.3B | - | 1 | 10.42 | 14.78 | 17.72 |
| GPTQ-3bit (Frantar et al., 2022) | 1.3B | - | 1 | 29.3 | 37.3 | 56.5 |
| GPTQ-2bit (Frantar et al., 2022) | 1.3B | - | 1 | 1.2e+6 | 1.3e+6 | 1.0e+6 |
| BiLLM (Huang et al., 2024a) | 1.3B | - | 1 | 4943.2 | 4013.6 | 3540.8 |
| Bi-Mamba (Tang et al., 2024b) | 1.3B | - | 1 | **12.6** | **13.6** | 28.9 |
| **SpikingMamba** | 1.3B | ✗ | $1 \times 1$ | 31.71 | 36.87 | 52.25 |
| **SpikingMamba** | 1.3B | ✗ | $1 \times 4$ | 17.90 | 23.45 | 30.19 |
| **SpikingMamba** | 1.3B | ✗ | $1 \times \pm 4$ | 15.79 | 21.01 | 26.20 |
| **SpikingMamba** | 1.3B | ✓ | $1 \times \pm 4$ | 15.17 | 20.66 | **25.44** |

## 5.3 Energy Analysis

SpikingMamba offers significant energy efficiency advantages over Mamba during inference. To quantify these benefits, we compare the energy efficiency ratio ($E_A/E_S$) between Mamba2 and SpikingMamba across different model sizes and neuron types (Additional details are provided in the Appendix C). As shown in Figure 2, larger models benefit more from spike-driven computation due to the proportion of linear projection increasing as the model scale. Across neuron types, LIF yields the highest efficiency due to its strict binarization, followed by I-LIF and SI-LIF. While SI-LIF slightly sacrifices efficiency, it enables significantly higher expressiveness and lower accuracy loss, highlighting a practical trade-off.

Introducing SGC (striped bars) slightly increases computation but reduces spike rates in most cases, especially in input projections, leading to an overall net energy gain. This effect holds even with SI-LIF neurons, where semantic fidelity is better preserved. Reinforcement learning (green bars) via DPO or KTO further improves accuracy without affecting energy use, since the architecture remains unchanged. Together, these results show that SGC and RL can be combined with SI-LIF to maintain high energy efficiency while narrowing the accuracy gap with the full Mamba2.

## 5.4 Ablation Study

We perform ablation studies to systematically evaluate the impact of key components in SpikingMamba, including the Neuron ablation, SGC integration, loss design and training framework. The results reveal which factors are critical for achieving competitive performance.

**SpikingMamba Architecture.** Table 5 shows that both Smoothed Gradient Compensation path and SI-LIF contribute significantly to performance. This ablation study on the 1.3B SpikingMamba model SI-LIF with $1 \times \pm 4$ and I-LIF $1 \times 4$). Removing the SGC path with a 0.76% performance drop. But replacing the SI-LIF with the I-LIF neuron results in a consistent drop in accuracy with 1.41%. Furthermore, SI-LIF best matches the Mamba2 block's output distribution, while LIF and I-LIF's positive-only spiking cause polarized outputs (Further details are provided in the Appendix D and E).

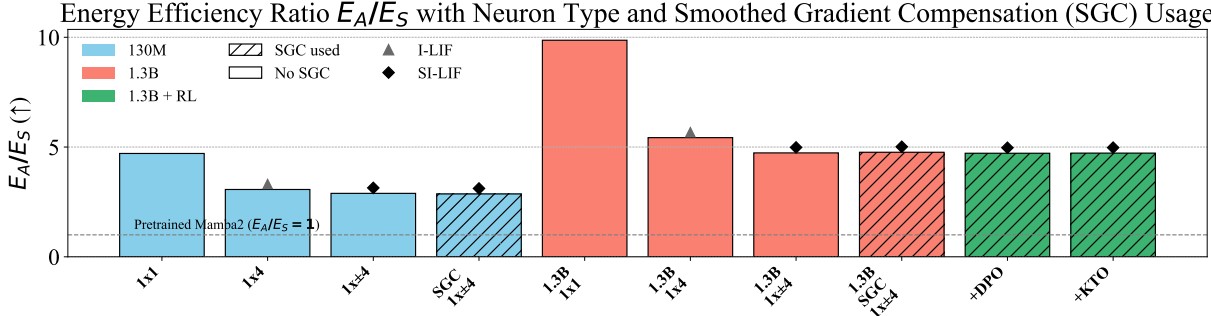

Figure 2: The inference energy efficiency ratio ($E_A/E_S$) of SpikingMamba under various configurations ($E_A$: energy of Mamba2, $E_S$: energy of SpikingMamba). Colors denote model size, striped bars indicate SGC path usage, and marker shapes indicate neuron types. Detailed data and fire rate are provided in Appendix C.

Table 5: Neuron ablation study on 1.3B Model.

| Method | SGC | Neuron | Avg. (%) | Diff. |
|---|---|---|---|---|
| SpikingMamba-1.3B | ✓ | SI-LIF | **54.62** | - |
| SpikingMamba-1.3B | ✗ | SI-LIF | 53.86 | -0.76 |
| SpikingMamba-1.3B | ✗ | I-LIF | 53.21 | -1.41 |
| SpikingMamba-1.3B | ✗ | LIF | 48.35 | -6.27 |

Table 6: Ablation study on the distillation loss.

| Method | $\mathcal{L}_{KL}$ | SGC | Avg. | Diff. |
|---|---|---|---|---|
| SpikingMamba-130M | ✓ | ✓ | 44.47 | - |
| SpikingMamba-130M w/o SGC | ✓ | ✗ | 44.09 | -0.38 |
| SpikingMamba-1.3B | ✓ | ✓ | 54.62 | - |
| SpikingMamba-1.3B w/o SGC | ✓ | ✗ | 53.86 | -0.76 |

**Distillation Loss.** Table 6 presents an ablation study on the distillation losses, highlighting that both components are essential. The combination of $\mathcal{L}_{KL}$ and SGC ($\mathcal{L}_{Hidden}$) yields the best results, while omitting either loss harms performance. For the 130M and 1.3B models, omitting SGC ($\mathcal{L}_{Hidden}$) resulted in accuracy degradation of 0.38% and 0.76%, respectively. This scaling trend suggests that larger models benefit more significantly from hidden-state alignment loss.

**Energy Accuracy Tradeoff.** Figure 3 illustrates how varying the integer spike range affects the energy–accuracy balance in both the 130M and 1.3B model configurations. To ensure fair comparison, all results in Figure 3 exclude the SGC module, which is specifically designed for SI-LIF and would otherwise confer an inherent advantage. Expanding the neuron range $D$, either by increasing its magnitude or by introducing symmetric negative values, consistently leads to higher accuracy while

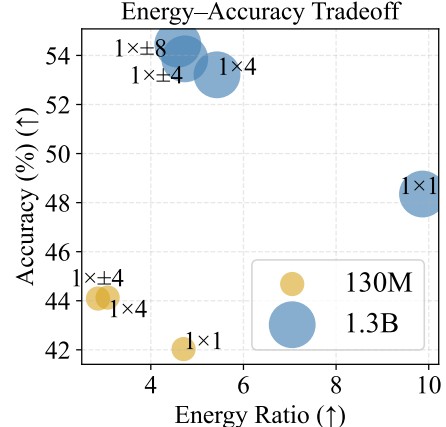

Figure 3: The Energy-accuracy tradeoff for 130M and 1.3B models under different spike neurons. Marker size indicates model scale.

simultaneously reducing the energy ratio ($E_{ANN}/E_{SNN}$). The trend is even more pronounced in the 1.3B model, indicating that broader integer spike ranges enhance representational capacity without compromising energy efficiency. Note that increasing $D$ improves accuracy, but it also introduces additional latency during inference.

**Training Framework w/o SGC Path.** Notably, the Smoothed Gradient Compensation (SGC) path delivers substantial performance gains despite minimal implementation, introduced in merely three layers. While the framework maintains functionality without SGC, evidenced by a moderate accuracy decline from 54.62% to 53.86%, its integration reveals significant synergistic effects with reinforcement learning. Crucially, KTO preference optimization achieves 57.17% accuracy with SGC versus 56.73% without, demonstrating a 0.44% absolute improvement. This confirms SGC's role as a gradient compensation mechanism that enhances reward-driven adaptation during alignment tuning.

# 6 Conclusion

We propose SpikingMamba for enabling energy-efficient inference of LLMs. By replacing dense projections with sparse spiking operations and compensating with a lightweight Smoothed Gradient Compensation path, SpikingMamba significantly reduces computation. A simple distillation and alignment strategy allows adaptation from a pretrained model without full retraining. Extensive experiments demonstrate that SpikingMamba achieves substantial energy efficiency improvements with minimal performance degradation across common language reasoning benchmarks.

**Acknowledgments**

This work is supported in part by the Science and Technology Innovation 2030-Major Project (Brain Science and Brain-Like Intelligence Technology) under Grant 2022ZD0208700, the National Key Research and Development Program of China (2021YFF1200800), the National Natural Science Foundation of China (Grant No.62276121, 12326604), Young Scientists Fund of the National Natural Science Foundation of China (Grant 62305278), The Guangdong Basic and Applied Basic Research Foundation (NO.2025A1515011758) and Youth S&T Talent Support Programme of Guangdong Provincial Association for Science and Technology (SKXRC2025460).

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

## A  Mamba2 Block

The Mamba2 (Dao & Gu, 2024) architecture consists of $L$ stacked layers. At each layer, given the input $\boldsymbol{u}_t \in \mathbb{R}^D$ at time step $t$, the processing begins with a unified input projection:

$$\boldsymbol{u}_t' = \boldsymbol{u}_t \boldsymbol{W}_{\text{in}} \in \mathbb{R}^{(2H \cdot P + 2N + H)}, \tag{16}$$

$$\boldsymbol{z}_t, \boldsymbol{x}_t', \boldsymbol{B}_t', \boldsymbol{C}_t', \Delta_t' = \text{Split}(\boldsymbol{u}_t'), \tag{17}$$

where $\boldsymbol{W}_{\text{in}} \in \mathbb{R}^{D \times (2H \cdot P + 2N + H)}$ is the input linear projection, $D$ is the model dimension, and $H, P, N$ are the number of heads, dimension of each head and state dimension, respectively. The hyperparameter often keeps the relationship with $H \cdot P = 2D$. The projected features $\boldsymbol{u}_t'$ are then split across heads and along channels. We reshape the activation to obtain input $\boldsymbol{x}_t'^{(d)} \in \mathbb{R}^P$, and $\Delta_t'^{(d)} \in \mathbb{R}$ for each head $d = 1, \dots, H$, The input-dependent variables for each head are computed as:

$$\alpha_t^{(d)} = \exp(-\Delta_t^{(d)} \exp(A^{(d)})) \in \mathbb{R},$$
$$\boldsymbol{C}_t^{(d)} = \sigma(\text{Conv1d}(\boldsymbol{C}_t')) \in \mathbb{R}^N,$$
$$\boldsymbol{B}_t^{(d)} = \sigma(\text{Conv1d}(\boldsymbol{B}_t')) \in \mathbb{R}^N,$$
$$\boldsymbol{x}_t^{(d)} = \sigma(\text{Conv1d}(\boldsymbol{x}_t'^{(d)})) \in \mathbb{R}^P,$$

where $\Delta_t^{(d)} = \text{Softplus}(\Delta_t'^{(d)} + \Delta_{\text{bias}}^{(d)}) \in \mathbb{R}$, and $\sigma(\cdot)$ is the SiLU activation and Conv1d denotes short causal convolution. The hidden state update and per-head output are computed as:

$$\boldsymbol{h}_t^{(d)} = \boldsymbol{h}_{t-1}^{(d)}(\alpha_t^{(d)} I) + (\Delta_t^{(d)} \boldsymbol{B}_t^{(d)}) \otimes \boldsymbol{x}_t^{(d)} \in \mathbb{R}^{N \times P}, \tag{18}$$

$$\boldsymbol{o}_t^{(d)} = \boldsymbol{C}_t^{(d)} \boldsymbol{h}_t^{(d)} + \boldsymbol{D}^{(d)} \odot \boldsymbol{x}_t^{(d)} \in \mathbb{R}^P, \tag{19}$$

$$\boldsymbol{y}_t = \text{Norm}\left(\text{Concat}(\boldsymbol{o}_t^{(1)}, \dots, \boldsymbol{o}_t^{(H)}) \odot z_t\right) \in \mathbb{R}^{H \cdot P}, \tag{20}$$

$$\boldsymbol{y}_t' = \boldsymbol{y}_t \boldsymbol{W}_{\text{out}} \in \mathbb{R}^D, \tag{21}$$

where $\otimes$ denotes the outer product, $\boldsymbol{D}^{(d)} \in \mathbb{R}^P$ and $\Delta^{(d)} \in \mathbb{R}$ are trainable parameters, Norm$(\cdot)$ denotes RMS normalization (Zhang & Sennrich, 2019), and $\boldsymbol{W}_{\text{out}} \in \mathbb{R}^{H \cdot P \times D}$ is the output projection matrix.

## B  Activation Statistics

To better understand the internal activation behavior of Mamba2, we visualize the distribution and structure of intermediate activations in Figures 4 and 5, and quantitatively evaluate their impact on model performance in Table 7.

Figures 4 show the layer-wise distributions of $\boldsymbol{u}_t$ and $\boldsymbol{y}_t$ across the 24-layer Mamba2-130M model. While $\boldsymbol{u}_t$ values are relatively centered and stable across layers, $\boldsymbol{y}_t$ exhibits increasingly high-magnitude activation in deeper layers, with a wide dynamic range. Figure 5 further confirms this by visualizing the token-wise and channel-wise activation values, where $\boldsymbol{y}_t$ shows sparse but extreme peaks.

To simulate spike-based activations, we clamp the maximum activation value per channel to either 0 or 1, mimicking a spiking output. Table 7 shows that this binarization leads to significant performance degradation across all tasks up to a 9.7% drop in average accuracy, demonstrating that high-magnitude activations play a critical role in preserving semantic fidelity. Notably, both $\boldsymbol{u}_t$ and $\boldsymbol{y}_t$ contribute to this effect, though the impact of clamping $\boldsymbol{y}_t$ is more severe due to its broader variance.

These results highlight a key challenge in deploying spike-based LLMs: naive binarization of positive activations leads to substantial loss in representational capacity, motivating our design of ternary neurons and auxiliary paths to preserve high-value semantics.

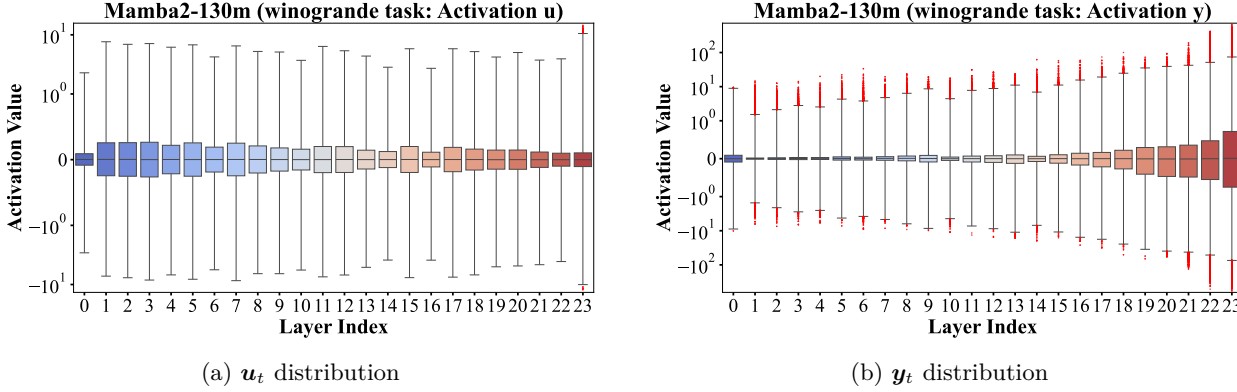

(a) $\boldsymbol{u}_t$ distribution

(b) $\boldsymbol{y}_t$ distribution

Figure 4: Activation distributions in Mamba2: (a) input projection $\boldsymbol{u}_t$ (Eq 16), (b) output projection $\boldsymbol{y}_t$ (Eq 21)

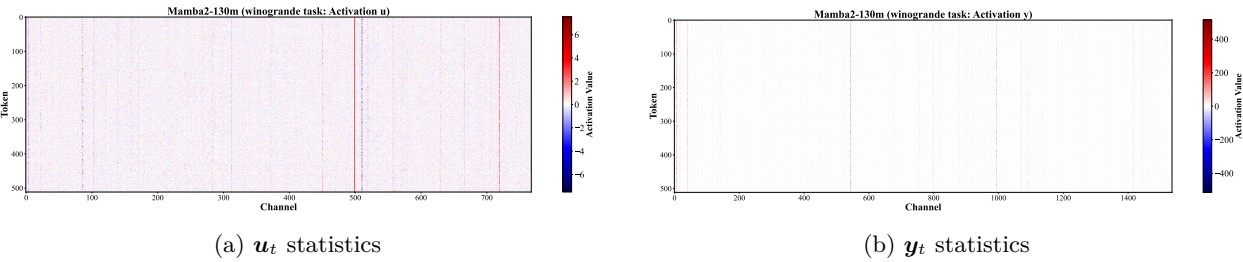

(a) $\boldsymbol{u}_t$ statistics

(b) $\boldsymbol{y}_t$ statistics

Figure 5: Activation statistics across channels and tokens in Mamba2: (a) input projection $\boldsymbol{u}_t$ (Eq 16), (b) output projection $\boldsymbol{y}_t$ (Eq 21).

## C  Energy Computation

### C.1  Energy Details

To quantify the energy efficiency of SpikingMamba compared to the original Mamba2 model, we present detailed energy consumption statistics for all layers in both the 130M and 1.3B model variants. Table 8 shows the total energy cost across major components such as input/output projection and SSM (state space model) for the 130M model. Similarly, Table 9 presents results for the 1.3B model, where we include settings with and without Smoothed Gradient Compensation (SGC) path and reinforcement learning (DPO/KTO). The detailed computation ref from Section C.2.

The energy is computed using a breakdown of core components, with the total cost measured in arbitrary units that are consistent across model types. The $E_\mathrm{A}/E_\mathrm{S}$ ratio in the final column captures the relative energy efficiency, where $E_\mathrm{A}$ represents the energy of the original Mamba2 model, and $E_\mathrm{S}$ corresponds to SpikingMamba under each variant. Notably, SpikingMamba configurations consistently reduce energy by 2.9–4.7× in the 130M model and up to 4.7-9.8× in the 1.3B model.

Table 7: Zero-shot accuracy obtained by setting the maximum activation in each channel to 1 or 0.

|  | HellaSwag | PiQA | Arc-E | Arc-C | BoolQ | WinoGrande | Avg. (%) | Diff. (%) |
|---|---|---|---|---|---|---|---|---|
| **Mamba2-130m** | 35.22 | 64.25 | 47.31 | 24.06 | 54.62 | 52.25 | 46.29 | - |
| $y_{\max} = 0$ | 27.93 | 53.16 | 31.44 | 24.15 | 40.18 | 51.54 | 38.07 | -8.22 |
| $y_{\max} = 1$ | 25.84 | 53.37 | 27.48 | 23.72 | 37.83 | 52.80 | 36.84 | -9.45 |
| $u_{\max} = 0$ | 26.18 | 50.60 | 26.22 | 25.34 | 40.24 | 50.83 | 36.57 | -9.72 |
| $u_{\max} = 1$ | 24.50 | 49.56 | 25.93 | 27.47 | 49.27 | 49.80 | 37.76 | -8.53 |

Table 8: The total energy consumption of the 130M model. The I-LIF is set as $T \times D = 1 \times 4$, SI-LIF is set as $T \times D = 1 \times \pm 4$. "SGC" means whether to use a smoothed gradient compensation path during training.

| | SGC | Step | $fr_{\text{in}}$ | $fr_{\text{out}}$ | In Proj. | Out Proj. | SSM | Others | Total ($\downarrow$) | $\frac{E_{\text{A}}}{E_{\text{S}}}$ ($\uparrow$) |
|---|---|---|---|---|---|---|---|---|---|---|
| **Mamba2** | - | 1 | - | - | 284.2067 | 130.2331 | 82.7476 | 1.4733 | 498.6607 | 1 |
| **SpikingMamba** | ✗ | $1 \times 1$ | 0.3180 | 0.1583 | 17.6826 | 4.0259 | 82.7476 | 1.4733 | 105.9294 | 4.7075 |
| **SpikingMamba** | ✗ | $1 \times 4$ | 0.3294 | 0.0509 | 73.1770 | 5.1878 | 82.7476 | 1.4733 | 162.5858 | 3.0671 |
| **SpikingMamba** | ✗ | $1 \times \pm 4$ | 0.3498 | 0.1215 | 77.8034 | 12.3835 | 82.7476 | 1.4733 | 174.4078 | 2.8592 |
| **SpikingMamba** | ✓ | $1 \times \pm 4$ | 0.3476 | 0.1236 | 77.3141 | 12.5975 | 82.7476 | 1.4733 | 174.1325 | 2.8637 |

Table 9: The total energy consumption of the 1.3B model. The I-LIF is set as $T \times D = 1 \times 4$, SI-LIF is set as $T \times D = 1 \times \pm 4$. "SGC" means whether to use a smoothed gradient compensation path during training.

| | SGC | Step | $fr_{\text{in}}$ | $fr_{\text{out}}$ | In Proj. | Out Proj. | SSM | Others | Total ($\downarrow$) | $\frac{E_{\text{A}}}{E_{\text{S}}}$ ($\uparrow$) |
|---|---|---|---|---|---|---|---|---|---|---|
| **Mamba2** | - | 1 | - | - | 3849.1128 | 1852.2046 | 441.3204 | 7.4809 | 6150.1188 | 1 |
| **SpikingMamba** | ✗ | $1 \times 1$ | 0.1605 | 0.1483 | 120.8705 | 53.7421 | 441.3204 | 7.4809 | 623.4140 | 9.8652 |
| **SpikingMamba** | ✗ | $1 \times 4$ | 0.2196 | 0.0156 | 661.5119 | 22.6130 | 441.3204 | 7.4809 | 1132.9263 | 5.4285 |
| **SpikingMamba** | ✗ | $1 \times \pm 4$ | 0.2529 | 0.0612 | 761.8833 | 88.6691 | 441.3204 | 7.4809 | 1299.3588 | 4.7332 |
| **SpikingMamba** | ✓ | $1 \times \pm 4$ | 0.2499 | 0.0619 | 752.7860 | 89.7272 | 441.3204 | 7.4809 | 1291.3147 | 4.7627 |
| **+ DPO** | ✓ | $1 \times \pm 4$ | 0.2533 | 0.0633 | 760.0157 | 93.0612 | 441.3204 | 7.4809 | 1301.8783 | 4.7240 |
| **+ KTO** | ✓ | $1 \times \pm 4$ | 0.2523 | 0.0624 | 763.0280 | 91.7566 | 441.3204 | 7.4809 | 1303.5860 | 4.7178 |

Table 10: Energy Evaluation for Mamba2 block.

| Operation | # Operations $\times$ # Energy |
|---|---|
| In Proj. | $(4D + 2N + H) \times D \times E_{\text{MM}}$ |
| Out Proj. | $D \times 2D \times E_{\text{MM}}$ |
| $dtB$ | $H \times ((P \times 1) \times (1 \times N)) \times E_{\text{MM}}$ |
| $C * h_t$ | $H \times (P \times N) \times E_{\text{MM}}$ |
| Conv1d | $(2D + 2N) \times 4 \times E_{\text{MM}}$ |
| Act | $3 \times (2D) \times E_{\text{EM}}$ |
| $x_t D$ | $H \times P \times E_{\text{EM}}$ |
| $xdB$ | $H \times P \times N \times E_{\text{EM}}$ |
| $Adt$ | $H \times E_{\text{EM}}$ |
| $Ah_t$ | $H \times P \times N \times E_{\text{EM}}$ |
| Norm | $2D \times E_{\text{EM}}$ |
| $y * act(z)$ | $2D \times E_{\text{EM}}$ |
| $dt + dt_{bias}$ | $H \times E_{\text{ADD}}$ |
| $Ah_t + xdB$ | $H \times P \times N \times E_{\text{ADD}}$ |
| $y + Dx$ | $H \times P \times E_{\text{ADD}}$ |

## C.2 Detail Computation

Tables 10–12 provide the exact computational formulations used for energy estimation, the operation mainly refers to Appendix A. These are derived based on the number of operations per module and their associated energy costs for different operation types. All operations assume a 32-bit floating-point implementation on 45nm technology, where matrix multiplication $E_{\text{MM}} = 4.6pJ$, element-wise multiplication $E_{\text{EM}} = 3.7pJ$ and addition $E_{\text{ADD}} = 0.9pJ$ (Horowitz, 2014).

Table 10 details the computation of energy consumption for Mamba2, listing core operations in each block such as projections, state transitions, and nonlinear components. Table 11 extends this to SpikingMamba using LIF neurons, where the energy is scaled by spike rates ($fr$), SGC path and spike-based neuron op-

Table 11: Energy Evaluation for SpikingMamba with LIF.

| Operation | # Operations $\times$ # Energy |
|---|---|
| In Proj. | $fr_{\text{in}} \times (4D + 2N + H) \times D \times E_{\text{ADD}}$ |
| Out Proj. | $fr_{\text{out}} \times D \times 2D \times E_{\text{ADD}}$ |
| $dtB$ | $H \times ((P \times 1) \times (1 \times N)) \times E_{\text{MM}}$ |
| $C * h_t$ | $H \times (P \times N) \times E_{\text{MM}}$ |
| Conv1d | $(2D + 2N) \times 4 \times E_{\text{MM}}$ |
| Act | $3 \times (2D) \times E_{\text{EM}}$ |
| $xD$ | $H \times P \times E_{\text{EM}}$ |
| $xdB$ | $H \times P \times N \times E_{\text{EM}}$ |
| $Adt$ | $H \times E_{\text{EM}}$ |
| $Ah_t$ | $H \times P \times N \times E_{\text{EM}}$ |
| Norm | $2D \times E_{\text{EM}}$ |
| $y * act(z)$ | $2D \times E_{\text{EM}}$ |
| $dt + dt_{bias}$ | $H \times E_{\text{ADD}}$ |
| $Ah_t + xdB$ | $H \times P \times N \times E_{\text{ADD}}$ |
| $y + Dx$ | $H \times P \times E_{\text{ADD}}$ |
| Spiking Neuron1 | $(D + D) \times E_{\text{EM}} + (D + D) \times E_{\text{ADD}}$ |
| Spiking Neuron2 | $(2D + 2D) \times E_{\text{EM}} + (2D + 2D) \times E_{\text{ADD}}$ |

Table 12: Energy Evaluation for SpikingMamba with I-LIF and SI-LIF, where $fr_{\text{in}} = \frac{\#\text{Spike}_{\text{in}}}{T \times k \times D}$, $fr_{\text{out}} = \frac{\#\text{Spike}_{\text{out}}}{T \times k \times 2D}$.

| Operation | # Operations $\times$ # Energy |
|---|---|
| In Proj. | $k \times fr_{\text{in}} \times (4D + 2N + H) \times D \times E_{\text{ADD}}$ |
| Out Proj. | $k \times fr_{\text{out}} \times D \times 2D \times E_{\text{ADD}}$ |
| $dtB$ | $H \times ((P \times 1) \times (1 \times N)) \times E_{\text{MM}}$ |
| $C * h_t$ | $H \times (P \times N) \times E_{\text{MM}}$ |
| Conv1d | $(2D + 2N) \times 4 \times E_{\text{MM}}$ |
| Act | $3 \times (2D) \times E_{\text{EM}}$ |
| $xD$ | $H \times P \times E_{\text{EM}}$ |
| $xdB$ | $H \times P \times N \times E_{\text{EM}}$ |
| $Adt$ | $H \times E_{\text{EM}}$ |
| $Ah_t$ | $H \times P \times N \times E_{\text{EM}}$ |
| Norm | $2D \times E_{\text{EM}}$ |
| $y * act(z)$ | $2D \times E_{\text{EM}}$ |
| $dt + dt_{bias}$ | $H \times E_{\text{ADD}}$ |
| $Ah_t + xdB$ | $H \times P \times N \times E_{\text{ADD}}$ |
| $y + Dx$ | $H \times P \times E_{\text{ADD}}$ |

erations are accounted for explicitly. Table 12 generalizes further to SpikingMamba with I-LIF and SI-LIF neurons. Here, separate spike rates $fr_{\text{in}}$ and $fr_{\text{out}}$ are introduced to model the proportion of spike activations relative to dense inputs/outputs, allowing precise modeling of spiking sparsity effects.

Together, these tables form the analytical backbone of the energy evaluations reported in the main text and demonstrate how sparsity and low-rank approximations contribute to overall energy reduction in Spiking-Mamba.

## D   Experimental Details

Table 13 presents the detailed zero-shot accuracy values and perplexity corresponding to the results reported in the main text.

Table 13: The energy consumption with different configurations. "P." means the model parameters. Step means $T \times D$ for Spiking Neuron. The standard LIF is $1 \times 1$, when $D > 0$ means using I-LIF, while $\pm D$ means using SI-LIF.

| | P. | r | Step | Total Energy (↓) | $\frac{E_A}{E_S}$ (↑) | Wiki2 (↓) | C4 (↓) | PTB (↓) |
|---|---|---|---|---|---|---|---|---|
| **Mamba2** | 130M | - | 1 | 498.6607 | 1 | 20.04 | 25.23 | 35.11 |
| **SpikingMamba** | 130M | ✗ | $1 \times 1$ | 105.9294 | 4.7075 | 55.76 | 55.97 | 89.74 |
| **SpikingMamba** | 130M | ✗ | $1 \times 4$ | 162.5858 | 3.0671 | 32.12 | 36.70 | 54.36 |
| **SpikingMamba** | 130M | ✗ | $1 \times \pm4$ | 172.5421 | 2.8901 | 28.55 | 33.62 | 48.26 |
| **Mamba2** | 1.3B | - | 1 | 6150.1188 | 1 | 10.42 | 14.78 | 17.72 |
| **SpikingMamba** | 1.3B | ✗ | $1 \times 1$ | 623.4140 | 9.8652 | 31.71 | 36.87 | 52.25 |
| **SpikingMamba** | 1.3B | ✗ | $1 \times 4$ | 1132.9263 | 5.4285 | 17.90 | 23.45 | 30.19 |
| **SpikingMamba** | 1.3B | ✗ | $1 \times \pm4$ | 1299.3538 | 4.7332 | 15.79 | 21.01 | 26.20 |
| GPTQ-3bit | 1.3B | - | 1 | N/A | N/A | 29.3 | 37.3 | 56.5 |
| GPTQ-2bit | 1.3B | - | 1 | N/A | N/A | 1.2e+6 | 1.3e+6 | 1.0e+6 |
| BiLLM | 1.3B | - | 1 | N/A | N/A | 4943.2 | 4013.6 | 3540.8 |
| Bi-Mamba | 1.3B | - | 1 | N/A | N/A | **12.6** | **13.6** | 28.9 |

## E   Block Activation Distribution Analysis

To further analyze the impact of different spiking neuron configurations, we visualize the activation distributions of hidden outputs from the final block across multiple 130M models. Figures 6–8 show histograms of activation values (x-axis) and their frequency (y-axis) for three model groups, each compared against the same pretrained baseline (Pretrained Mamba2).

In Figure 6, the LIF model exhibits severe output polarization due to its binary firing nature. Figure 7 demonstrates that I-LIF reduces this binarization effect through continuous activations. However, its positive-only outputs still lead to a skewed distribution. Finally, Figure 8 shows that SI-LIF, which supports both positive and negative outputs, resolves the polarization issue entirely.

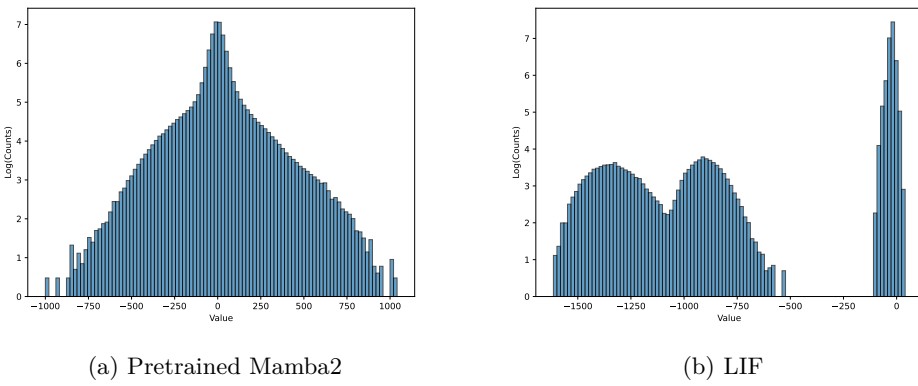

(a) Pretrained Mamba2                     (b) LIF

Figure 6: Activation distribution comparison for LIF-based models.

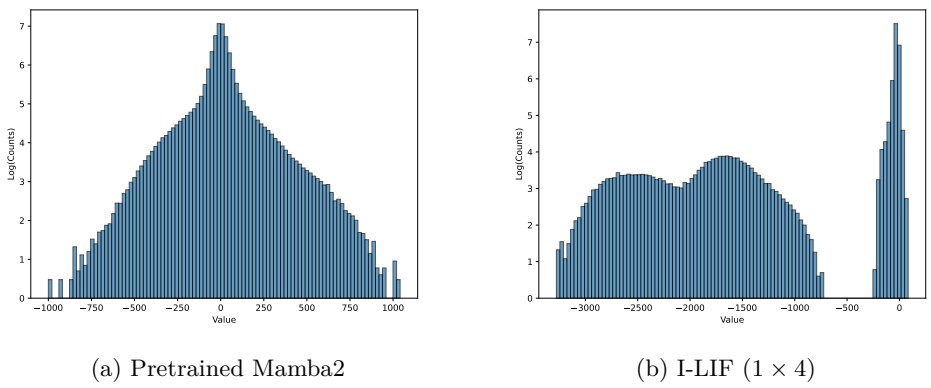

(a) Pretrained Mamba2           (b) I-LIF (1 × 4)

Figure 7: Activation distribution comparison for I-LIF (1 × 4) models.

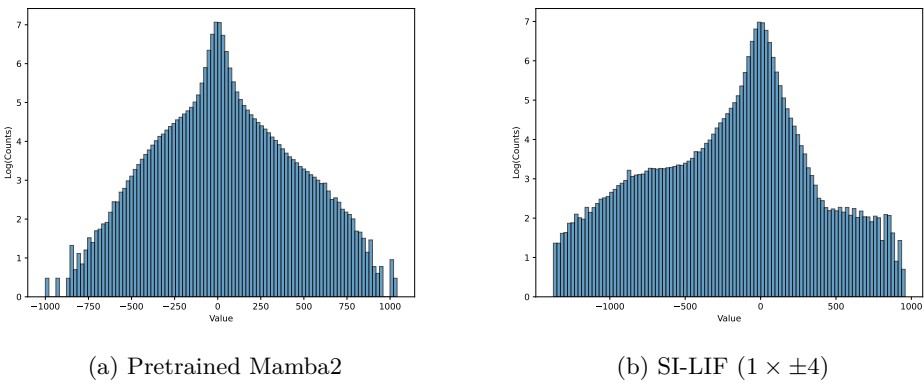

(a) Pretrained Mamba2           (b) SI-LIF (1 × ±4)

Figure 8: Activation distribution comparison for SI-LIF (1 × ±4) models.

