# OpenReview forum: "SpikingMamba: Towards Energy-Efficient Large Language Models via Knowledge Distillation from Mamba"
_TMLR — Accepted by TMLR_

### Review · Reviewer_xvRP · 2025-11-06

**Summary Of Contributions:**

This paper proposes SpikingMamba, an energy-efficient SNN-based LLMs distilled from Mamba2 that improves energy efficiency with minimal accuracy sacrifice. TI-LIF is a ternary-integer spiking neuron that extends the integer LIF neuron. Smoothed Gradient Compensation (SGC) is developed to alleviate the gradient approximation errors during back-propagation due to the quantized characteristics of TI-LIF neurons. Finally, self-distillation and reinforcement learning algorithms (DPO and KTO) are adopted to transfer the zero-shot ability from Mamba2 and to achieve preference alignment. Experiments demonstrate that SpikingMamba obtains energy efficiency improvements with minimal performance degradation across multiple language reasoning datasets, outshining previous spiking based language models.

**Additional Comments:**

Minor points:

1.	Abstract: “achieves a further 2.55% accuracy improvement after RL”. This sentence is somewhat misleading. At first glance I thought that using RL algorithms the performance can outperform the original Mamba2. The fact is that the results after RL are still inferior to Mamba2.


2.	A host of references listed in the introduction make the reader difficult to read the main idea of this paper. The authors should cite several references as needed after the sentence or some notable research lines, rather than give a long list of references.


3.	The introduction section should focus on the main idea of this paper, and defer other irrelevant works into the Related Work section. For example, the discussion on KV-cache, quantization for LLMs, and ANN-to-SNN conversion, are irrelevant to the main work of this paper.


4.	Page 2, the full name of SSM should be offered before using the abbreviation.


5.	Page 2, bold fonts are misused too much in the contribution part.


6.	Table 1, the notation D should be explained in the table title.


7.	Page 4, eq. (1) and (2). Notations should be given before the LIF neuron model, otherwise the reader feels hard to understand. Also, I want to know the input and output of this model. I read the reference (Wu 2018) and found that the notations and formula are very different to eq. (1) and (2) in this paper. Hence it is strange to cite Wu 2018. Why not cite the original LIF paper or some significant survey paper in spiking networks?


8.	Page 4, what is the meaning of “sparse AC operations”?


9.	Page 4, eq. (4). The y[t] is only related to W, not with x. The original formula is y[t]=Wx[t].


10.	Page 5, eq. (6). When x_t is in [-D, D], why is the derivative equal to alpha rather than 1? How can we handle the round operator when calculating the derivative?


11.	Page 5, eq. (7) and (8). I can not understand that beta=1 can ensure exact reconstruction. Why, and what to reconstruct?

12.	Page 6, eq. (10). What is the implication of “semantic consistency”?

13.	Table 2. The T times D notation follows the paper Yao 2025. Since the authors have read that paper, it is curious to ask “why the method proposed in Yao 2025 is not compared in Table 2 and following tables?”

14.	Page 8, table 3. SGC does not occur in this table, so there is no need to describe SGC.

15.	Page 8, Perplexity. Table 4 should be referred in the main text.

16.	Page 9, Energy Analysis. Is this part of results devoted to training or inference?

**Audience:**

Yes

**Audience Explanation:**

Spiking neural networks can provide an energy-efficient alternative to large language models such as Mamba2.

**Broader Impact Concerns:**

I think there is no need to add a Broader Impact Statement.

**Claims And Evidence:**

Yes

**Claims Explanation:**

The claims made in the submission are supported by several experimental results.

**Requested Changes:**

Pros:

1.	Although I am not working in spiking neural networks in recent years, it appears that the claims made in the submission are supported by convincing and clear evidence.


2.	The main contributions and the technical details are clearly presented, despite some contents requiring more explanations (see minor points below).


Cons:

1.	The three contributions summarized in the introduction sections appears to be straightforward and do not exhibit strong novelty. TI-LIF neurons simply incorporate -1 into the existing integer LIF neurons. Smoothed Gradient Compensation is a good idea to decrease the approximation errors, but the technical complexity is low. The third contribution, single-stage self-distillation and the use of reinforcement learning algorithms, seems to be weak and less convincing.


2.	The presentation could be improved. For instance, the writing style (with bold headlines) in Introduction section is not common in academic papers. Also there are some claims that needs more explanations (see below).

---

> ### Author Response · Authors · 2025-11-19
> **Reply (1/3)**
>
> > **Cons1:** The three contributions summarized in the introduction sections appears to be straightforward and do not exhibit strong novelty. TI-LIF neurons simply incorporate -1 into the existing integer LIF neurons. Smoothed Gradient Compensation is a good idea to decrease the approximation errors, but the technical complexity is low. The third contribution, single-stage self-distillation and the use of reinforcement learning algorithms, seems to be weak and less convincing.
>
> **Reply Cons1:** Thank you for this valuable and constructive review. Our motivation is to address the dense-computation bottleneck in existing Mamba models by exploring an energy-efficient SNN alternative through knowledge distillation. However, distilling an ANN teacher into an SNN student faces two fundamental challenges: (1) severe feature quantization caused by discrete spikes, and (2) gradient mismatch arising from nondifferentiable neuronal dynamics. To address these issues, our work integrates three components: (1) SI-LIF for stable integer activations, (2) Smoothed Gradient Compensation to reduce approximation error, and (3) a reinforcement learning stage to further close the teacher–student performance gap.
>
> This combined design proves effective in practice, as demonstrated by the 1.3B model, where the distilled SNN achieves competitive accuracy with substantial energy savings.
>
> ---
> > **Cons2:** The presentation could be improved. For instance, the writing style (with bold headlines) in Introduction section is not common in academic papers. Also there are some claims that needs more explanations (see below).
>
> **Reply Cons2:** Thank you for the constructive suggestion. We have revised the writing style in the Introduction section by removing the bold headlines and adopting a more standard academic layout.
>
> ---
> > **1:** Abstract: “achieves a further 2.55% accuracy improvement after RL”. This sentence is somewhat misleading. At first glance I thought that using RL algorithms the performance can outperform the original Mamba2. The fact is that the results after RL are still inferior to Mamba2.
>
> **Reply Comment 1:** Thank you for pointing out this ambiguity.  Our intended meaning was that the reinforcement learning (RL) stage provides an additional 2.55% improvement over our own baseline (before RL fine-tuning), not over Mamba2. To avoid misunderstanding, we have revised the sentence in the Abstract to: “The model achieves a further 2.55% accuracy improvement after RL, narrowing the performance gap from 4.78% to 2.23%.”
>
> We appreciate your constructive suggestion; the revisions are highlighted in blue in the abstract of the revised manuscript.
>
> ---
> > **2:** A host of references listed in the introduction make the reader difficult to read the main idea of this paper. The authors should cite several references as needed after the sentence or some notable research lines, rather than give a long list of references.
>
> **Reply Comment 2:** Thank you for the helpful suggestion. All references have been reorganized and updated accordingly in the revised manuscript.
>
> ---
> > **3:** The introduction section should focus on the main idea of this paper, and defer other irrelevant works into the Related Work section. For example, the discussion on KV-cache, quantization for LLMs, and ANN-to-SNN conversion, are irrelevant to the main work of this paper.
>
> **Reply Comment 3:** Thank you for the helpful suggestion. To improve the coherence of the Introduction, we have reorganized the structure by retaining only the background and motivation relevant to this work and moving the discussions on quantization and ANN-to-SNN conversion to the Related Work section.
>
> The corresponding revisions are highlighted in blue on Pages 1-2 of the updated manuscript.
>
> ---
> > **4:** Page 2, the full name of SSM should be offered before using the abbreviation.
>
> > **5:** Page 2, bold fonts are misused too much in the contribution part.
>
> > **6:** Table 1, the notation D should be explained in the table title.
>
> **Reply Comments 4-6:** We appreciate the reviewer’s careful reading and constructive feedback. To address these comments: (1) The full term State Space Model (SSM) has been introduced before its first occurrence on Page 1 to ensure clarity and consistency. (2) The use of bold fonts in the contribution section has been refined to follow standard academic formatting and improve readability. (3) The notation $D$ has been explicitly defined in the caption of Table 1 for clarity.
>
> All corresponding revisions have been incorporated into the manuscript and are highlighted in blue in the revised version on Pages 1 and 2.

---

> > ### Author Response · Authors · 2025-11-19
> > **Reply (2/3)**
> >
> > > **7:** Page 4, eq. (1) and (2). Notations should be given before the LIF neuron model, otherwise the reader feels hard to understand. ...  Why not cite the original LIF paper or some significant survey paper in spiking networks?
> >
> > **Reply Comments 7:** Thank you for this constructive suggestion. To clearly define all variables and improve readability, We have added a dedicated Notation paragraph before introducing the LIF neuron equations on Page 3.
> >
> > Additionally, we have revised the references accordingly: instead of citing Wu (2018), we now cite more appropriate papers on the Leaky Integrate-and-Fire (LIF) model [1-2] to ensure accuracy and consistency with standard formulations.
> >
> > All corresponding revisions are incorporated into the revised manuscript and highlighted in blue.
> >
> > [1] CLIF: complementary leaky integrate-and-fire neuron for spiking neural networks[C]//Proceedings of the 41st International Conference on Machine Learning. 2024: 19949-19972.
> >
> > [2] Towards memory-and time-efficient backpropagation for training spiking neural networks[C]//Proceedings of the IEEE/CVF international conference on computer vision. 2023: 6166-6176.
> >
> > ---
> >
> > > **8:** Page 4, what is the meaning of “sparse AC operations”?
> >
> > **Reply Comments 8:** Thank you for your valuable question. “Sparse AC operations’’ refer to Accumulation (AC) operations without the multiplication step. This occurs in event-driven spiking computation, where additions are performed only when spikes are present, therefore leading to sparsity in the operation count. We will add the full term “Accumulation (AC)” at its first occurrence in the revised manuscript to avoid ambiguity.
> >
> > We have added the above description of sparse AC to the motivation section of the revised manuscript on Page 4.
> >
> > ---
> > > **9:** Page 4, eq. (4). The y[t] is only related to W, not with x. The original formula is y[t]=Wx[t].
> >
> > **Reply Comments 9:** Thank you for your constructive question. For original Eq.(4) is
> >
> > $\boldsymbol{y}[t] = \sum_{\\{ i \mid \boldsymbol{s}[t]_i = 1\\}} \boldsymbol{W}\_{:,i}$,
> >
> > where we should explain more clearly in the main text. Specifically, the conditional summation is determined by the sparse binary activation$\boldsymbol{s}[t] = f_{\text{SN}}(\boldsymbol{x}[t]) \in \\{0,1\\}^{d_{\text{in}}}$. Where the spike $\boldsymbol{s}[t]$ serves as condition, generated by the spike neuron $f_{\text{SN}}(\cdot)$ from the input $\boldsymbol{x}[t]$.
> >
> > We have inserted the clarification directly following Eq. (4) on Page 4, and the updated text is highlighted in blue in the revised manuscripts.
> >
> > ---
> >
> > > **10:** Page 5, eq. (6). When x_t is in [-D, D], why is the derivative equal to alpha rather than 1? How can we handle the round operator when calculating the derivative?
> >
> > **Reply Comment 10:** Thank you for pointing out this issue. In our formulation, the coefficient $\alpha$ denotes the slope of the surrogate gradient. In practice, we use alpha=1 by default without further tuning.
> >
> > Regarding the round operator, we handle its non-differentiability using the straight-through estimator (STE). Specifically, the rounding function is implemented as:
> >
> > ${\rm STE}(\boldsymbol{x}_t)= (\operatorname{clamp}(\operatorname{round}(\boldsymbol{x}_t),-D,D) -\operatorname{clamp}(\boldsymbol{x}_t,-D, D)).{\rm detach()}+\operatorname{clamp}(\boldsymbol{x}_t,-D,D) $,
> >
> > which ensures that the forward pass uses the discrete rounded value, while the backward pass propagates the gradient through the continuous term. This allows us to retain discrete spike representations in the forward computation while keeping the gradient flow stable during training.
> >
> > We have added an explicit explanation after Eq. (6) to clarify the role of $\alpha$ and how the rounding operator is handled through the surrogate gradient. The updated text is highlighted in blue in the revised manuscript on Page 5.

---

> > > ### Author Response · Authors · 2025-11-19
> > > **Reply (3/3)**
> > >
> > > ---
> > > > **11:** Page 5, eq. (7) and (8). I can not understand that beta=1 can ensure exact reconstruction. Why, and what to reconstruct?
> > >
> > > **Reply Comments 11:** Thank you for this thoughtful and constructive question. During spike-driven inference, the integer activation $\boldsymbol{s}_t$ is reconstructed by emitting $\boldsymbol{s}[i]$ spikes across $D$ micro-timesteps and then applying the sign flag. The total number of emitted spikes exactly matches $|\boldsymbol{s}_t|$, and the sign bit recovers its polarity. This reconstructed spike count selects the corresponding rows of $\boldsymbol{W}$, yielding a computation that is exactly equivalent to the dense operation $\boldsymbol{W}\boldsymbol{s}_t$.
> > >
> > > Setting $\beta=1$ removes membrane decay, effectively turning the LIF neuron into an integrate-and-fire neuron without leakage. In this case, the membrane potential accumulates linearly, ensuring that the number of emitted spikes matches the intended integer activation, thereby enabling exact reconstruction.
> > >
> > > For clarity, we have added a notation summary on Page 3 and updated Eq. (9) on Page 5 to better illustrate how SI-LIF dynamics achieve this spike-based equivalence. These revisions are highlighted in blue in the updated manuscript.
> > >
> > > ---
> > > > **12:** Page 6, eq. (10). What is the implication of “semantic consistency”?
> > >
> > > **Reply Comments 12:** Thank you for raising this constructive question. In our manuscripts, “semantic consistency’’ refers to aligning the functional outputs of the spiking pathway and the SGC pathway so that they represent similar predictive semantics at each timestep.
> > >
> > > We have clarified this definition in the revised manuscript on Page 6 with highlighted blue color.
> > >
> > > ---
> > > > **13:** Table 2. The T times D notation follows the paper Yao 2025. Since the authors have read that paper, it is curious to ask “why the method proposed in Yao 2025 is not compared in Table 2 and following tables?”
> > >
> > > **Reply Comments 13:** Thank you for your constructive question. When $D>1$, i.e., in the case of $T \times D = 1 \times 4$ for I-LIF, the setting corresponds exactly to the neuron proposed in Yao (2025). To avoid misleading, we will add the corresponding citation after “I-LIF” in the table.
> > >
> > > We have also updated the captions of Table 2 and Table 4 to explicitly clarify that the $T \times D = 1 \times 4$ case follows Yao (2025).
> > >
> > > ---
> > > > **14:** Page 8, table 3. SGC does not occur in this table, so there is no need to describe SGC.
> > >
> > > > **15:** Page 8, Perplexity. Table 4 should be referred in the main text.
> > >
> > > > **16:** Page 9, Energy Analysis. Is this part of results devoted to training or inference?
> > >
> > > **Reply Comments 14-16:** Thank you for the careful reading and helpful suggestions. (1) The mention of SGC has been removed from Table 3 as suggested. (2) Table 4 is now properly referenced in the main text within the Perplexity Analysis paragraph. (3) The Energy Analysis section refers to inference energy rather than training energy. To avoid ambiguity, we have explicitly revised the wording to “inference energy” in the updated manuscript.
> > >
> > > All corresponding modifications are highlighted in blue in the revised Manuscript Page 9-10.
> > >
> > > ---
> > > Finally, we sincerely thank you again for your meticulous review. The constructive comments you provided have been highly valuable in strengthening the manuscript. Should there be any further questions, we would be more than willing to respond and make additional improvements.

---

### Review · Reviewer_QixS · 2025-11-08

**Summary Of Contributions:**

The authors propose to distill mamba2 in a partially spiking neural net architecture which is modeled closely to the original mamba2 architecture. From a high level view, they introduce spiking computations in the input projection layer and in the output projection layer of a mamba block. They are justifying this choice of partial spike-introduction.
They propose distillation as training mode which makes sense.
As for additional technical contributions, they propose to use during training a ternary integer-valued lif neuron, a temporary non-spiking block ensuring an additional gradient flow path called Smoothed Gradient Compensation.

They also evaluate the impact of DPO and KTO reinforcement learning for post-distillation alignment.

In terms of presentation, they show a comparison to other spiking architectures in relevant parameters in Table 1 which is informative and an overview graphic in Figure 1.

They measure zero-shot accuracy on LLM benchmark tasks.
They measure zero-shot perplexity on two textual tasks for comparison against other spiking models.

An energy analysis is performed with details given in the appendix.

**Additional Comments:**

None

**Audience:**

Yes

**Audience Explanation:**

- The paper proposes a partially spiking neural net architecture which may hold promise for more energy-efficient inference. It does this for Mamba, which is a reasonable competitor of attention-based LLMs.

- They provide it with improvements of the gradient flow for distillation training.

- The model achieves an acceptable tradeoff between accuracy loss and energy-reduction .

- The paper is also in many parts well readable. Current readability issues (see requested changes) is something that can be healed in a revision.

Overall it is of high interest for parts of the community and is in the view of this reviewer a good paper.

**Claims And Evidence:**

Yes

**Claims Explanation:**

The claims are very well supported.
 They show a slight gap to the original Mamba2 with a gain in energy efficiency. They show experimental results for the gap, and bring a computation of the energy efficiency based on the observed operations. They claim two technical contributions and discuss them well in the paper. They show the impact of the proposed changes with an ablation study. They measure zero-shot perplexity on LLM tasks. The claims are well supported by the experiments.

**Requested Changes:**

The paragraph Spike-Driven Inference is hard to read:

- The authors do not write at its start what they aim to do.  It seems that they want to compute $y_t = W x_t$ using a spiking architecture where typically $x_t \in [-D,+D]$. please correct if something is half wrong in this claim. Whatever is correct, please write what you want to do in this paragraph.

- With the preceding section one expect $s_t$ to be used in there. An additional sentence, that during training $x_t$ will be replaced by $s_t$ would be good (if that is not mistaken by me).

- equation (9): there is something confusing about sign(x_t): Should it not be component-wise multiplied to s[i] (as it is input to the matrix multiplication with W) ? That is to put it inside the sums and as $ \mathrm{sign}(\mathbf{x}_t)_j s[i]_j  $ ?

- Table 1 should name D as integer range

- "Upon completion of training, only **this** spiking pathway is retained for event-driven
inference."

change to: Upon completion of training, only **the** spiking pathway is retained for event-driveninference.

- It is not a must-change, however, ternary integer LIF is a bit misleading, though without doing harm: Ternary comes from three. With a general D this is gone. It would be better to name it signed-integer LIF (SI-LIF).

---

> ### Author Response · Authors · 2025-11-19
>
> > The paragraph Spike-Driven Inference is hard to read:
>
> > The authors do not write at its start what they aim to do. ..., please write what you want to do in this paragraph.
>
> **Reply 1:** Thank you for this constructive and helpful comment. The goal of the “Spike-Driven Inference’’ paragraph is to show how the dense operation $\boldsymbol{y}_t = \boldsymbol{W}\boldsymbol{s}_t$ is implemented exactly through sparse spike-based accumulations. It explains how the trained SI-LIF neuron reconstructs the integer activation $\boldsymbol{s}_t = \text{SI-LIF}(\boldsymbol{x}_t)$, and how a spiking micro-timestep process can compute the same transformation $\boldsymbol{y}_t = \boldsymbol{W}\boldsymbol{s}_t$ using event-driven binary spikes rather than dense MAC operations.
>
> All clarifications have been added to the revised manuscript, with the new text highlighted in blue on Page 5.
>
> ---
> > With the preceding section one expect  to be used in there. An additional sentence, that during training will be replaced by would be good (if that is not mistaken by me).
>
> **Reply 2:** Thank you for the helpful comment. To avoid confusion and ensure consistent notation, we have added a notation summary on Page 3 and an explicit clarifying sentence on Page 5, both highlighted in blue in the revised manuscript.
>
> ---
> > equation (9): there is something confusing about sign(x_t): Should it not be component-wise multiplied to s[i] (as it is input to the matrix multiplication with W) ? That is to put it inside the sums and as $\text{sign}(\boldsymbol{x}_t)_j \boldsymbol{s}[i]_j$
>
> **Reply 3:** Thank you for the insightful comment. You are correct that $\mathrm{sign}(\boldsymbol{x}_t)$ should be applied component-wise inside the summations. We have revised Eq.(9) accordingly by placing the sign factor within the sums as $\mathrm{sign}(\boldsymbol{x}_t)_j s[i]_j$ for making the expression clearer and mathematically consistent.
>
> The updated equation is highlighted in blue in the revised manuscript on Page 5.
>
> ---
> > Table 1 should name D as integer range
>
> **Reply 4:** Thank you for the helpful suggestion. We have ensured consistent usage of the notations $D$ throughout the paper. The definitions have been added to the caption of Table 1, and the corresponding revisions are highlighted in blue in the updated manuscript.
>
> ---
> > "Upon completion of training, only this spiking pathway is retained for event-driven inference." change to: Upon completion of training, only the spiking pathway is retained for event-driveninference.
>
> **Reply 5:** Thank you for the helpful comment. We have modified this sentence on Page 6 of the revised manuscript.
>
> ---
> > It is not a must-change, however, ternary integer LIF is a bit misleading, though without doing harm: Ternary comes from three. With a general D this is gone. It would be better to name it signed-integer LIF (SI-LIF).
>
> **Reply 6:** Thank you for the constructive suggestion. To meet your comments, we have updated the terminology to use signed-integer LIF (SI-LIF) consistently throughout the revised manuscript.
>
> ---
>
> Finally, we sincerely thank you again for your meticulous review. The constructive comments you provided have been highly valuable in strengthening the manuscript. Should there be any further questions, we would be more than willing to respond and make additional improvements.

---

### Review · Reviewer_y4Kx · 2025-11-11

**Summary Of Contributions:**

In the interest of implementing large language models more efficiently, there have been various recent proposals for using sparsely spiking neurons (i.e. those with quantized outputs) to construct models for which inference is faster and more efficient. This paper contributes a novel method for constructing a model which uses spiking neurons at key bottlenecks, based on the Mamba2 architecture [Dao & Gu, 2024]. Specifically, the paper's novel insights include a quantization scheme which allows spiking neurons to take a range of positive and negative values ("Ternary Integer Leaky-Integrate-and-Fire") as well as the use of an additional, fully-differentiable path at training time to reduce quantization issues while computing gradients ("Smoothed Gradient Compensation Path"). The method's effectiveness is evaluated by comparing its accuracy as a language model against the Mamba2 base model plus various other spiking models, plus its energy efficiency against Mamba2, and through an ablation study.


Tri Dao and Albert Gu. "Transformers are SSMs: Generalized models and efficient algorithms through structured state space duality." arXiv preprint arXiv:2405.21060, 2024

**Additional Comments:**

* It is stated that SGC is only used for three layers: the "first", "last", and "middle". I'm not sure which these are.
* The spiking neural networks community has surely explored neurons similar to the TI-LIF neurons examined here, right? It would be useful to mention this literature or else comment upon the gap.

**Audience:**

Yes

**Audience Explanation:**

I expect this paper to be of some immediate interest to researchers working on efficient language models. It may also find some readers in the broader spiking neural networks community for whom the demonstrated performance advantage of TI-LIF spiking neurons at scale is relevant.

**Broader Impact Concerns:**

It would be good to provide a comprehensive summary of resources used in generating the experimental data.

**Claims And Evidence:**

Yes

**Claims Explanation:**

I found the experimental evaluation to be reasonably comprehensive and convincing. One shortcoming is that the ablation study demonstrates that increasing the range of integer values for spikes improves performance; however, only two ranges are examined, so it would be good to explore this trend a little more thoroughly. It would also be useful to establish the energy/accuracy tradeoff a little more rigorously, perhaps as a scatterplot with energy and accuracy as axes.

**Requested Changes:**

* Please keep the usage of the notation $T$ and $D$ consistent and define them both before they are used in Table 1.
* The evaluation protocol should be defined more clearly in the main text. What exactly does zero-shot accuracy mean here? What data are used to estimate it?

---

> ### Author Response · Authors · 2025-11-19
>
> > One shortcoming is that the ablation study demonstrates that increasing the range of integer values for spikes improves performance; however, only two ranges are examined, so it would be good to explore this trend a little more thoroughly. It would also be useful to establish the energy/accuracy tradeoff a little more rigorously, perhaps as a scatterplot with energy and accuracy as axes.
>
> **Reply 1:** Thank you for the constructive suggestion. We have expanded the ablation study by including additional integer ranges and added a scatter plot illustrating the energy/accuracy tradeoff. We extended the range to $\pm8$ and observed a performance gain. However, this also reduced energy efficiency and doubled the inference latency.
>
> The new figure and analysis have been incorporated into the ablation study section and highlighted in blue in the revised manuscript on Page 10.
>
> ---
>
> > Please keep the usage of the notation $T$ and $D$ consistent and define them both before they are used in Table 1.
>
>
> **Reply 2:** Thank you for the helpful suggestion. We have ensured consistent usage of the notations $T$ and $D$ throughout the paper.
>
> The definitions have been added to the caption of Table 1, and the corresponding revisions are highlighted in blue in the updated manuscript.
>
> ---
>
> > The evaluation protocol should be defined more clearly in the main text. What exactly does zero-shot accuracy mean here? What data are used to estimate it?
>
> > It would be good to provide a comprehensive summary of resources used in generating the experimental data.
>
> **Reply 3:** We sincerely thank you for this constructive comment. We have clarified the evaluation protocol in the main text. The relevant descriptions and dataset details have been moved from the Appendix to Section 5.1 and are highlighted in blue in the revised manuscripts.
>
> Our evaluation strictly follows the standard LM-Eval-Harness protocol, which has been widely adopted in prior work on large language models (e.g., GPT, LLaMA, Mamba) for reproducible zero-shot assessment across benchmarks such as LAMBADA, PIQA, ARC and so on.
>
> Specifically, zero-shot accuracy measures a model’s performance on downstream tasks without any task-specific fine-tuning or gradient updates, i.e., the model is directly evaluated in an inference-only manner using task instructions and prompts.
>
> ---
>
> > It is stated that SGC is only used for three layers: the "first", "last", and "middle". I'm not sure which these are.
>
> **Reply 4:** Thank you for this valuable comment. For the Mamba2-130M model, which contains 24 layers, the “first,” “middle,” and “last” layers correspond to the 1st, 12th, and 24th layers, respectively. For the 1.3B model, which contains 48 layers, they correspond to the 1st, 24th, and 48th layers.
>
> We have clarified this in the revised manuscript on Page 7, and the corresponding update is highlighted in blue.
>
> ---
>
> > The spiking neural networks community has surely explored neurons similar to the TI-LIF neurons examined here, right? It would be useful to mention this literature or else comment upon the gap.
>
> **Reply 5:** Thank you for the insightful comment. Indeed, neurons similar to the TI-LIF/I-LIF family have been explored in the SNN community. Expect discussed in section 3.1, concurrent work SpikingBrain [1] mainly focuses on integer-based quantized inference and does not verify end-to-end gradient-based training for the spiking model, as their quantization mechanisms are designed for forward pass efficiency rather than backward optimization.
>
> To clarify this connection and distinguish our SI-LIF neuron from prior quantized LIF variants, we have added a brief discussion after Section 3.1, with the new text highlighted in blue in the revised manuscript on Page 4.
>
> ---
> **Reference:**
>
> [1] Pan Y, Feng Y, Zhuang J, et al. SpikingBrain: Spiking Brain-inspired Large Models[J]. arXiv preprint arXiv:2509.05276, 2025.
>
> ---
> Finally, we sincerely thank you again for your meticulous review. The constructive comments you provided have been highly valuable in strengthening the manuscript. Should there be any further questions, we would be more than willing to respond and make additional improvements.

---

### Decision · Action_Editor_k4oo · 2026-01-12

**Recommendation:** Accept with minor revision

**Additional Comments:**

I thank the authors for their submission to TMLR.  I am requesting very minor writing changes, in line with the earlier request from Reviewer QixS:
- Could the authors please remove the bold words preceding each paragraph in the Introduction?  In the case of the first instance, **Large Language Models**, this can just be unbolded.  The paper flows well without these lead-in phrases and, as Reviewer QixS, such bolded phrases are uncommon in the introduction and can hurt the readable of the paper when overused
- For Section 2, please consider the following changes: a) Making "Mamba and Distillation" a subsection, (b) replacing "SNNs and quantization for efficient LLMs" with a subsection "SNNs for LLMs and Mamba", (c) removing bolded phrase "SNNs for Language Modeling." and "SNNs with Mamba."
- Changing "3.1 Spiking Neuron" to "3.1 Spiking Neurons"
- Removing the following bolded phrases from Section 3: "I-LIF Neuron."
- Replacing the bolded phrases preceding paragraphs in subsection 4.1, 4.3, and 5.2 with subsubsections
- Note: the use of these bolded phrases looks good in 5.1 and 5.4

**Audience:**

Yes

**Audience Explanation:**

All reviewers agree the general TMLR audience would be interest in this work; Mamba models remain strong competitors to attention-based LLMs, particularly due to their efficient (sub-quadratic) training complexities while remaining competitive to Transformer architectures.  Thus, further improvements in efficiency (i.e., energy efficiency, as described in the paper) for Mamba would be of interest to TMLR's audience.

**Claims And Evidence:**

Yes

**Claims Explanation:**

All reviewers agree that the work applies a solid application of spiking neural networks to Mamba LLMs.  Furthermore, all reviewers agree that the majority of the claims were well supported, with requested experiments from Reviewer QixS added during the rebuttal.  The authors additionally addressed minor issues and clarifications.